# MeshCoder: LLM-Powered Structured Mesh Code Generation from Point Clouds

**Bingquan Dai**[*1,2], **Li Luo**[*1,3], **Qihong Tang**[1,4], **Jie Wang**[1,5], **Xinyu Lian**[1], **Hao Xu**[1,6], **Minghan Qin**[2], **Xudong Xu**[1], **Bo Dai**[3], **Haoqian Wang**[†2], **Zhaoyang Lyu**[†1], **Jiangmiao Pang**[1]

[1]Shanghai Artificial Intelligence Laboratory  [2]Tsinghua University  [3]The University of Hong Kong
[4]Harbin Institute of Technology  [5]Beijing Institute of Technology  [6]AI Thrust, HKUST(GZ)

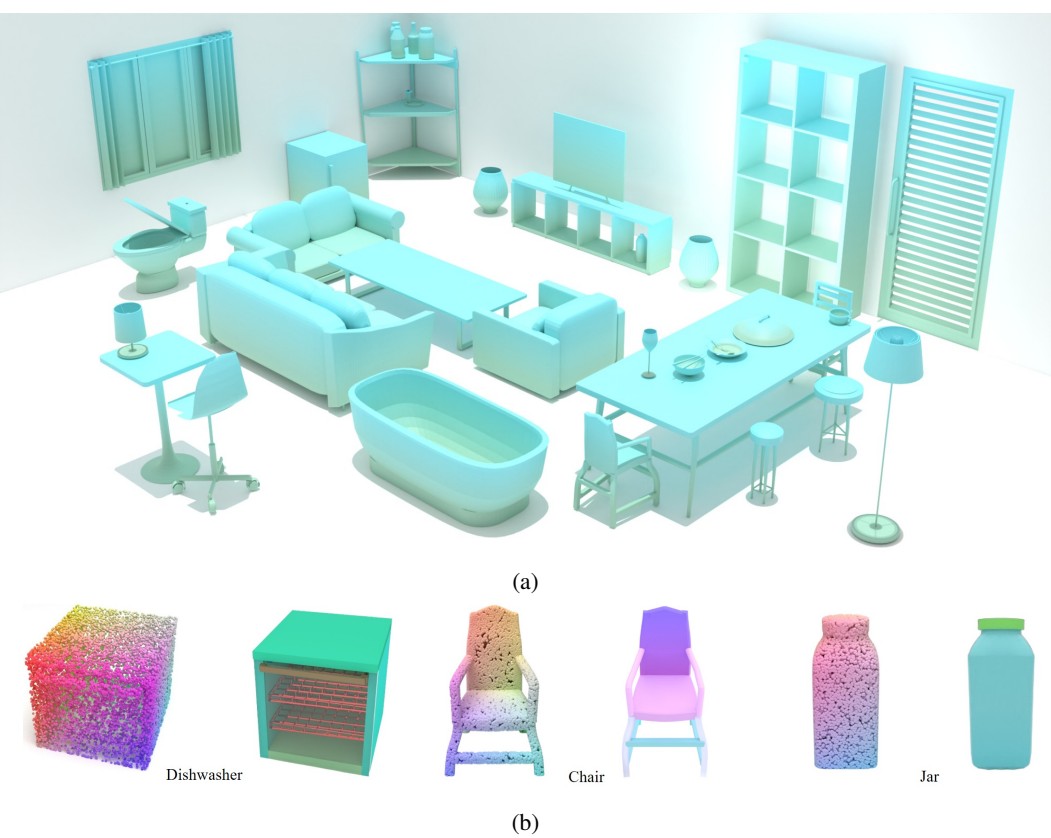

(a)

(b)

Figure 1: (a) MeshCoder can predict codes and reconstruct 41 categories of objects. (b) MeshCoder takes in point clouds and produce part-segmented meshes by executing the predicted code in Blender. For the dishwasher, we apply transparency to the foremost part to showcase the internal structure.

## Abstract

Reconstructing 3D objects into editable programs is pivotal for applications like reverse engineering and shape editing. However, existing methods often rely on limited domain-specific languages (DSLs) and small-scale datasets, restricting their ability to model complex geometries and structures. To address these challenges, we introduce MeshCoder, a novel framework that reconstructs complex

*Equal contribution.
†Corresponding authors.
 Project leader: Zhaoyang Lyu

39th Conference on Neural Information Processing Systems (NeurIPS 2025).

3D objects from point clouds into editable Blender Python scripts. We develop a comprehensive set of expressive Blender Python APIs capable of synthesizing intricate geometries. Leveraging these APIs, we construct a large-scale paired object-code dataset, where the code for each object is decomposed into distinct semantic parts. Subsequently, we train a multimodal large language model (LLM) that translates 3D point cloud into executable Blender Python scripts. Our approach not only achieves superior performance in shape-to-code reconstruction tasks but also facilitates intuitive geometric and topological editing through convenient code modifications. Furthermore, our code-based representation enhances the reasoning capabilities of LLMs in 3D shape understanding tasks. Together, these contributions establish MeshCoder as a powerful and flexible solution for programmatic 3D shape reconstruction and understanding. Project homepage: `https://daibingquan.github.io/MeshCoder`.

# 1   Introduction

Inferring shape programs from 3D observations is of great importance for reverse engineering, shape editing, and 3D structure understanding. Prior work [1, 2, 3] has explored this problem by defining Domain-Specific Languages (DSLs) to model geometric and structural properties of objects and training neural networks to map 3D observations to shape programs. However, existing methods struggle to generalize to objects with complex geometry and structure. Two key limitations underlie this gap. First, existing DSLs are constrained to modeling simple primitives (e.g., cubes, spheres, cylinders) and cannot represent real-world objects with intricate parts. Second, training shape-to-code inference models demands large-scale paired datasets of 3D objects and their corresponding code, while such datasets are scarce. Prior work often relies on datasets with limited categories, geometric complexity and part count.

To address these challenges, we introduce MeshCoder, a novel framework for generating Blender Python scripts that reconstruct complex 3D objects into their constituent parts. First, we design a set of expressive Blender Python APIs that are capable of synthesizing intricate geometries beyond simple primitives. For instance, our APIs can create complex shapes by translating a 2D section curve along a specified trajectory, bridging section curves of different shapes, adding bevels or applying Boolean operations on basic shapes, repeating a basic shape in one dimension or two dimensions. With these concise yet powerful Blender Python APIs, we can model highly complex shapes, addressing the limitations of prior DSLs.

Second, we present a novel pipeline to construct a large-scale paired object-code dataset. We begin by synthesizing diverse object parts using our APIs with parametrically sampled parameters, yielding a part-level dataset. A part-to-code inference model is then trained on this dataset to predict code for individual parts. Next, we employ this model to construct a holistic object-code dataset. We use Infinigen-Indoor [4] to generate a dataset of objects, and each object is decomposed into its constituent parts. We use the part-to-code inference model to predict code for each part of an object, and then carefully design rules to concatenate code of all parts to obtain code of the object. This process yields a dataset of approximately 1 million objects spanning 41 categories, with objects up to more than 100 parts. Finally, we train a multimodal large language model (LLM) on this dataset to infer code from 3D objects. We use point clouds as 3D shape representations due to their ease of acquisition, and use a triplane-based tokenizer to transform the input point cloud to a set of fixed-length tokens. These tokens are fed into the LLM to generate Blender Python scripts that replicate input geometries in distinct semantic parts.

We evaluate our approach against existing shape-to-code methods, with experimental results and quantitative metrics demonstrating that our framework significantly outperforms prior work. Furthermore, by representing shapes as executable code, our method facilitates intuitive geometric and topological editing through simple code modifications. This capability enables precise alterations to object geometry and mesh topology, enhancing flexibility in downstream applications. Additionally, we conduct experiments on shape structural and geometric understanding tasks, revealing that our code-based representation improves the reasoning capabilities of large language models (LLMs) when interpreting 3D shapes. In summary, our contributions are outlined as follows:

- We have developed a comprehensive set of Blender Python APIs, facilitating the modeling of intricate geometries. This enhanced API suite empowers the procedural generation of complex 3D structures, effectively addressing the limitations of traditional domain-specific languages (DSLs) in representing detailed and varied shapes.

- We propose a pipeline to construct a large-scale paired object-code dataset. Using the dataset we constructed, we can train an shape-to-code inference model.

- We trained MeshCoder, an **Object-to-Code inference framework** that generates Blender Python scripts to reconstruct 3D meshes from point clouds in a structured and editable manner. Our model encodes 3D shapes into part-level code, simplifying mesh editing and enhancing LLMs' understanding of 3D objects.

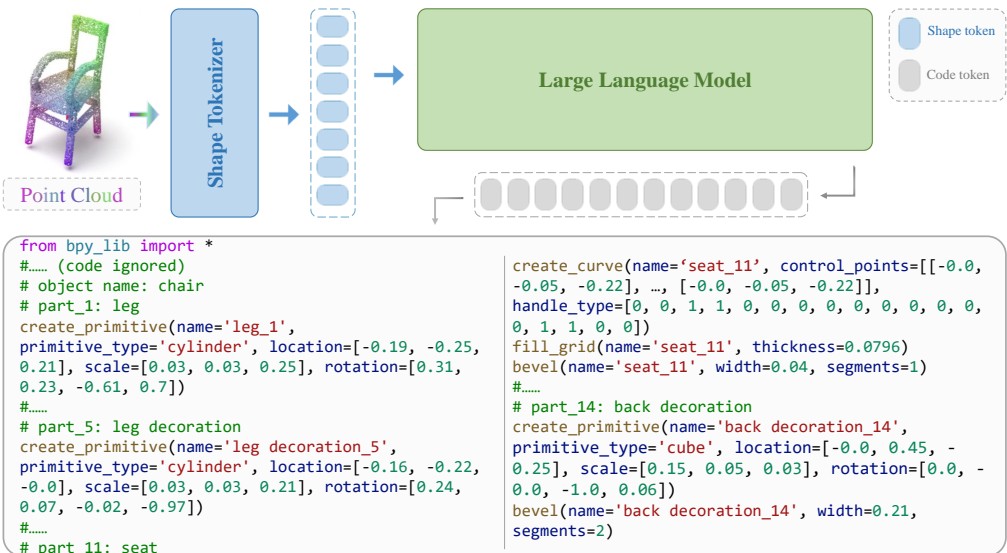

Figure 2: Overview of MeshCoder. The input point cloud is first encoded into shape tokens via a shape tokenizer. These tokens are then fed into a large language model (LLM), which autoregressively generates executable code representing part-based 3D structures. The decoded code specifies object's name, part identities and names, enabling interpretable and modular reconstruction.

## 2 Related Work

**Shape programs.** Shape programs provide a structured and interpretable framework for representing 3D geometry by utilizing domain-specific languages to describe the generative processes of shapes. Early work such as ShapeAssembly [5] introduced explicit shape programs that capture the hierarchical and part-based organization of objects. Subsequent methods, including ShapeCoder [6], PLAD [2], and ShapeLib [7], progressively improved program abstraction, learning efficiency, and scalability with large language models. Other approaches, such as those proposed by Liang [3] and Tian et al. [1], incorporate differentiable rendering or neuro-symbolic reasoning to enhance program inference and execution. While these methods exhibit strong generalization capabilities in composing simple geometric elements like boxes and cylinders, they often struggle to model complex part geometries or generate artist-grade quad meshes, which restricts their application in high-fidelity asset creation. In addition, a range of methods in CAD program generation [8, 9, 10, 11, 12, 13] have explored synthesizing code representations for individual CAD parts. However, these approaches are limited to isolated component generation and lack the capability to model complete multi-part objects with coherent structural relationships.

**Part-based Representation.** Part-based representations have proven highly valuable in 3D shape analysis and synthesis. Some approaches [14, 15, 16, 17, 18, 19, 20, 21, 22, 23] take a generative approach, assembling objects by combining predefined or learned parts into complete 3D structures. Other methods [24, 25, 26, 27, 28, 29, 30, 31, 32, 33, 34, 35] focus on segmenting 3D objects

into individual parts, enabling more modular and flexible manipulation of shapes. For instance, SAMPart3D [24] introduces a scalable zero-shot 3D part segmentation framework that segments any 3D object into semantic parts at multiple granularities without requiring predefined part label sets as text prompts. PartSLIP [28] explores low-shot part segmentation of 3D point clouds by leveraging a pretrained image-language model, GLIP, transferring rich knowledge from 2D to 3D through GLIP-based part detection on point cloud rendering and a novel 2D-to-3D label lifting algorithm. SATR [26] performs zero-shot 3D shape segmentation via text descriptions by using a zero-shot 2D object detector, inferring 3D segmentation from multi-view 2D bounding box predictions by exploiting the topological properties of the underlying surface. Despite these advancements in part segmentation and reconstruction, these methods do not translate segmented parts into executable code representations, limiting their integration into code-driven design workflows.

# 3 Methodology

As shown in Figure 2, we aim to train an object-to-code inference model that takes in a point cloud of an object, and then predict the Blender python scripts of each part of the object. When executing the python scripts in Blender, we can obtain the same object in separated parts. To train such an object-to-code inference model, we need a dataset of paired objects and the corresponding codes. To obtain such a dataset, we first train a part-to-code inference model that predicts code for a single part on our synthetic dataset of paired parts and the corresponding codes. Then, given a dataset of objects separated in different parts, we use the trained part-to-code inference model to predict code for every part of an object. Finally, we concatenate the codes of every part of the object and obtain the code of the object. Now, we have a dataset of paired objects and the corresponding codes, and are ready to train the object-to-code inference model.

We explain the key steps described above in details in the following sections. First, we explain how to synthesize a dataset of paired parts and the corresponding codes in Section 3.1. Then, we describe the training procedure of the part-to-code inference model in Section 3.2. Next, we use the part-to-code inference model to obtain the code of an entire object in Section 3.3. Finally, we train the object-to-code inference model in Section 3.4.

## 3.1 Part Dataset

We aim to generate a dataset of paired part shapes and codes. To do so, we implement probabilistic programs to generate Blender Python scripts, and obtain the corresponding shape by executing the code in Blender. We carefully design these probabilistic programs and ensure that the shapes generated are within the range $[-1, 1]^3$. There are several types of shapes that we generate, as illustrated in Figure 3. We explain them in the following paragraphs.

**Primitive.** Primitives are a set of fundamental geometric shapes, consistent with those defined in Blender. Specifically, we consider five basic shapes: `cube`, `cylinder`, `UV sphere`, `cone`, and `torus`. Each primitive is parameterized by three attributes: `location` (**location** $\in \mathbb{R}^3$), `rotation` (**rotation** $\in \mathbb{H}$), and `scale` (**scale** $\in \mathbb{R}^3$), where $\mathbb{H}$ denotes the space of unit quaternions. The `location` specifies the shape's position in 3D space, `rotation` defines its orientation via quaternions, and `scale` determines the shape's size along its local axes. Examples of Primitives can be found in the first row of Figure 3.

**Translation.** Translation is defined as the geometry obtained by sweeping a 2D cross-sectional shape along a 3D trajectory curve, which is equivalent to Sweep in CAD. As illustrated in the second row of Figure 3, during this translation process, the tangent direction of the 3D trajectory remains perpendicular to the 2D shape, and the size of the section shape can change along the 3D trajectory. For a more detailed explanation, please refer to A.1. To implement this, we first define a 2D shape using a set of control points (i.e., spatial coordinates), and then specify a 3D trajectory curve in a similar manner. Specifically, our experiments consider five types of cross-sectional shapes: rectangles, circles, circular arcs, polygons, and Bézier curves. For the trajectories, we define six forms: straight lines, polylines, circles, circular arcs, rectangles, and Bézier curves. Notably, this method also allows a 2D shape to rotate around an axis to form a solid of revolution, making it suitable for modeling objects such as bottles and plates.

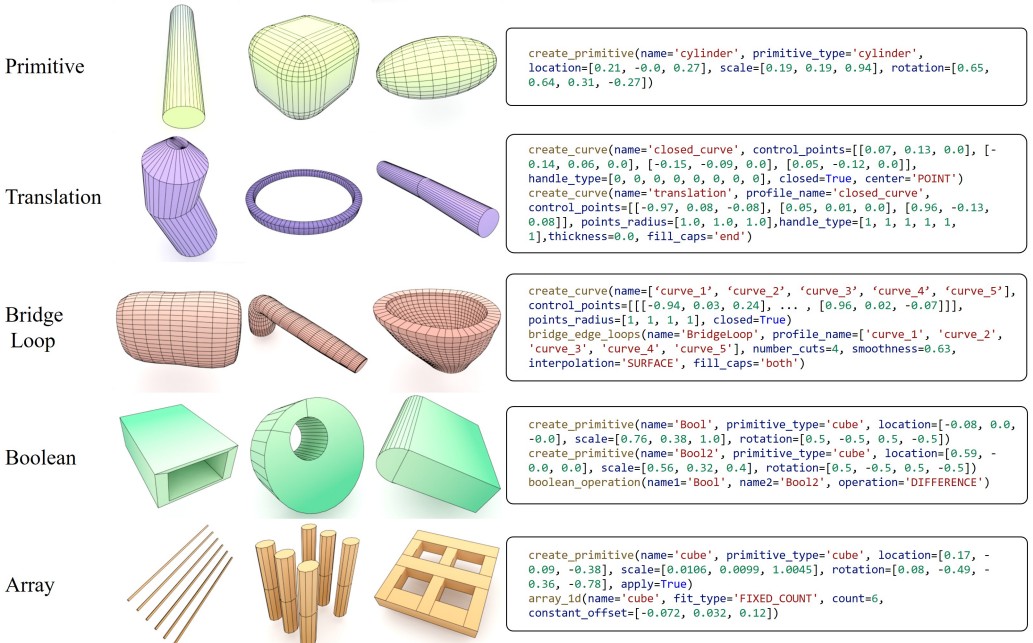

Figure 3: Visualization of basic geometric shape types and their corresponding code. For each shape category, the code shown corresponds to the first example.

**Bridge loop.** Although the Translation method is capable of generating certain complex objects, it remains constrained in several ways. For instance, in the Translation operation, the 2D cross-sectional shape is always orthogonal to the tangent direction of the trajectory. Moreover, the section shape is allowed to change only in scale, without any deformation in its geometry. To address this limitation, we introduce an alternative method for constructing geometries, namely the Bridge Loop. This geometry is constructed by first generating a sequence of 2D shapes and then connecting their corresponding vertices to form a continuous 3D geometry, which is equivalent to Loft in CAD. Some cases can be seen in the third row of Figure 3. The Bridge Loop approach enables the creation of more complex geometries compared to those achievable via Translation alone. For a more detailed explanation, please refer to A.1.

**Boolean.** Boolean geometries refer to geometries formed by applying Boolean operations—namely union, intersection, and difference—to two or more of the fundamental shape categories defined in Section 3.1. The union operation enables the construction of complex composite geometries, while the difference operation is used to generate geometries with holes or indentations. We can see some examples and their corresponding codes in the fourth row of Figure 3.

**Array.** When a particular type of primitive geometry appears repeatedly in a regular pattern, we do not invoke the construction function for each primitive individually, as this would result in lengthy code. Instead, we employ an Array method to construct the entire structure collectively. Specifically, we define two types of Arrays: 1D Arrays, where a geometry is repeatedly instantiated along a curve, and 2D Arrays, where repetition occurs across a plane. Cases of this type can be seen in the last row of Figure 3.

It is important to note that when designing our functions, the function parameters include two types: array-based parameters and individual parameters. During dataset construction, we select the parameter format based on the most suitable fit for the object: individual parameters are used for simpler parts, while array-based parameters are adopted when a part contains multiple similar shapes. For instance, the translation example in Figure 3 uses individual parameters, whereas the bridge loop example employs array-based parameters.

After designing the five fundamental types of template functions described above, we perform probabilistic random sampling over these functions and their parameters to generate a series of function code snippets. For each sampled code snippet $y$, we execute it to obtain the corresponding mesh $M$. In this way, we construct a dataset of paired code $y$ and mesh $M$.

## 3.2 Part-to-code Inference Model

After constructing the dataset of paired code $y$ and mesh $M$, we sample a point cloud $x \in \mathbb{R}^{N \times 3}$ from each mesh $M$, where $N$ is the number of points in the point cloud. We train a part-to-code inference model $h$ that takes in a point cloud $x$ and predict the corresponding code $y$. The inference model consists of two modules: The shape tokenizer model and a fintuned LLM. The tokenizer model takes in the point cloud $x$ and outputs a set of fixed length tokens $z \in \mathbb{R}^{L \times D}$, where $L$ is the number of shape tokens and $D$ is the dimension of each token. We set $D$ to the same dimension as the word embeddings in the LLM. Thereafter, the LLM takes in the shape tokens $z$ and then predict $y$, the code of the point cloud $x$. We train the shape tokenizer model and finetune the LLM at the same time using the cross-entropy loss for the prediction of the next token in the shape code $y$. We use Llama-3.2-1B as the base LLM and finetune it using LoRA.

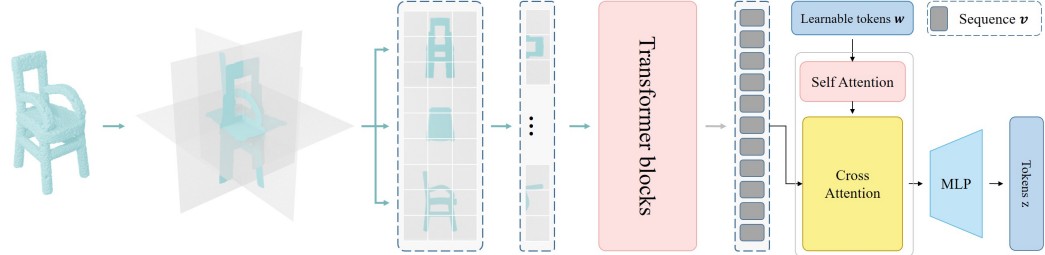

Figure 4: Architecture of the shape tokenizer. We first project the point cloud into the triplane and obtain triplane features. The triplane features are patchified and reshaped into a 1D sequence, and fed into transformer blocks to obtain triplane tokens. Finally, we use a set of learnable tokens to aggregate information from triplane tokens via cross-attention.

**The shape tokenizer model.** We explain the detailed structure of the shape tokenizer model. As shown in Figure 4, the shape tokenizer model transforms a point cloud $x \in \mathbb{R}^{N \times 3}$ to a set of fixed length tokens $z \in \mathbb{R}^{L \times D}$. We first project the point cloud $x$ to a triplane and obtain triplane feature $u \in \mathbb{R}^{3 \times H \times W \times D_1}$, where $H, W$ are the height and width of the planes, and $D_1$ is the dimension of the triplane feature. The coordinates of each point is fed to a shared MLP and a feature of dimension $D_1$ is obtained. We project each point's feature to the three perpendicular planes according to the point's position. Features projected to the same pixel are aggregated by max-pooling. Pixels that do not correspond to any point are filled with zeros. After obtaining the triplane feature $u$, we patchify it and reshape it into a $1D$ sequence $v \in \mathbb{R}^{(3 \cdot H/f \cdot W/f) \times D_1}$, where $f$ is the patch size. We then feed the sequence $v$ to a set of transformer blocks and outputs $v' \in \mathbb{R}^{(3 \cdot H/f \cdot W/f) \times D_1}$. Next, to compress the number of tokens fed into the LLM, we use a learnable set of tokens $w \in \mathbb{R}^{L \times D_2}$ to aggregate information from $v'$ using cross attention:

$$\text{CrossAttn}(\text{Transformer}(w), v', v'), \tag{1}$$

where Transformer denotes a transformer block, $\text{CrossAttn}(Q, K, V)$ denotes a cross attention block, and $Q, K, V$ are query, key, value, respectively. By feeding $w$ to a set of these cross attention blocks, we obtain tokens $w' \in \mathbb{R}^{L \times D_2}$ that contain information about the point cloud $x$. Finally, we use an MLP to transform the dimension of $w'$ from $D_2$ to $D$ and obtain shape tokens $z \in \mathbb{R}^{L \times D}$, where $D$ is the dimension of the word embeddings in the LLM. Now, the shape tokens $z$ can be readily fed to the LLM and predict the code corresponding to the point cloud $x$.

## 3.3 Assemble Parts to Objects

After training the part-to-code inference model $h$, we can use it to obtain the code of an object. Given a dataset of objects, in which each object $\mathcal{O}$ is separated into its constituent parts $\mathcal{O} = \{q_i | i = 1, 2, \cdots, M\}$, where $q_i$ is the $i$-th part of object $\mathcal{O}$, and $M$ is the number of parts of the object $\mathcal{O}$. We also assume that each part $q_i$ has its semantic label. We can use the part-to-code inference model $h$ to obtain the code of each part. Specifically, we first normalize each part $q_i$ to the cube $[-1, 1]^3$ according to its minimum bounding box and obtain the shape $q_i'$. Then we use the part-to-code inference model $h$ to obtain its code $y_i' = h(q_i')$. We then implement algorithms to transform the relevant numerical parameters in the code $y_i'$ to the original location, scale, and pose of $q_i'$ and obtain the code $y_i$ of the original shape $q_i$. Finally, we concatenate the codes of all parts of

the object, add semantic information to the code for each part, and obtain the code of the object $\boldsymbol{y} = \{\boldsymbol{y}_i | i = 1, 2, \cdots, M\}$. When concatenating the code, we sort each part based on its spatial position. Specifically, we assign an index to each part following a spatial order from bottom to top, left to right, and front to back. An overview of this pipeline is illustrated in Figure 5. During code inference, the part point cloud $\boldsymbol{q}_i$ is first transformed into a canonical space using a rotation matrix $\boldsymbol{R}$, translation matrix $\boldsymbol{T}$, and scaling factor $s$, resulting in $\boldsymbol{q}'_i$. The trained part-to-code inference model $\boldsymbol{h}$ generates the code $\boldsymbol{y}'_i$ of $\boldsymbol{q}'_i$. $\boldsymbol{y}'_i$ is then transformed back to the original pose and scale using the inverse of $\boldsymbol{R}, \boldsymbol{T}$, and $s$, and we obtain the code $\boldsymbol{y}_i$ of $\boldsymbol{q}_i$.

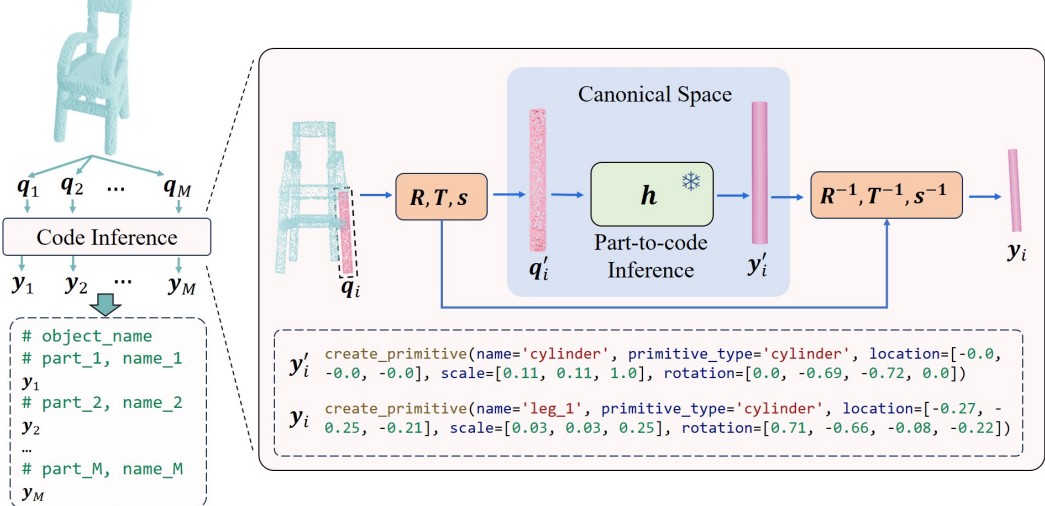

Figure 5: Pipeline of object-level code dataset construction using the part-to-code inference model. For each part point cloud $\boldsymbol{q}_i$, the code inference module independently predicts its corresponding code $\boldsymbol{y}_i$. All part codes $\boldsymbol{y}_i$ are then concatenated to form the complete object code. We also add meaningful semantic information to the object code following the template shown in the figure. The complete code of the example chair is shown in Figure 2.

## 3.4 Object-to-code Inference Model

After obtaining the code $\boldsymbol{y}$ of each object $\mathcal{O}$ in the dataset, we can use them to train an object-to-code inference model. Our object-to-code inference model has the same structure as the part-to-code inference model described in Section 3.2. We initialize the weights of the object-to-code inference model as the weights of the trained part-to-code inference model, and use the same training method in Section 3.2 to train the object-to-code inference model. It is worth noting semantic information in the ground-truth code of objects enables the object-to-code inference model to learn the semantic structure of objects, and facilitate 3D shape understanding.

## 4 Experiment

### 4.1 Datasets

#### 4.1.1 Synthetic Part Dataset

To facilitate the training of our part-to-inference model, we first constructed a synthetic part dataset. Specifically, we utilized functions from our basic shape code library, randomly sampling their parameters based on manually defined distributions to generate paired data of synthetic parts and corresponding code. This process yielded 1.5 million point cloud–code pairs for primitive shapes, 3 million for Translation-based parts, 1.5 million for Bridge Loop structures, 1.5 million for Boolean operations, and 2.4 million for Array-based constructions. In total, our constructed part dataset comprises around 10 million point cloud–code pairs. We partitioned the dataset into 70% for training, 15% for validation, and 15% for testing.

### 4.1.2   Object Dataset

We trained our model on the Infinigen Indoor [4] dataset. **Infinigen Indoor** is a procedural framework for generating synthetic 3D indoor objects, where each generated instance is automatically composed by its corresponding parts. We have made extensive modifications to the original Infinigen codebase to enable it to produce both individual components and their complete assemblies. Using this framework, we constructed a synthetic dataset comprising 41 common object categories, generating 1 million object-code pairs in total. We partitioned the dataset into training, validation, and test sets, following the same split strategy as the Synthetic Part Dataset. For more details, please refer to the A.1.

### 4.2   Implementation Details

We conduct training and evaluation on the Infinigen Indoor datasets [36]. For the part-to-code reconstruction model, we adopt the AdamW optimizer and train it for 20 epochs on NVIDIA A100 GPUs with a batch size of 512, and a learning rate of $10^{-4}$. We evaluate the model at every epoch and select the checkpoint with the lowest $L_2$ Chamfer Distance (CD) loss. Then we initialize the weights of the object-to-code reconstruction model with the weights of the trained part-to-code reconstruction model, and train the model on Infinigen Indoor dataset for 10 epochs, with a batch size of 256, and a learning rate of $10^{-4}$. The checkpoint with the lowest CD loss is selected. For additional training details and the parameter settings of the models, please refer to A.3 and A.2.

### 4.3   Reconstruction Performance

For reconstruction performance, we compare our method with two representative shape-to-code baselines, Shape2Prog [1] and PLAD [2]. Figure 6 illustrates visualization comparisons of results. We adopt IoU and $L_2$ CD as our evaluation metrics. Specifically, we voxelize the model's predicted outputs into $32^3$ grids and compute the IoU between the predicted and ground truth voxel grids. In parallel, we sample point clouds from both the predicted outputs and the ground truth, and calculate the Chamfer Distance between the two point clouds. Regarding the number of points and normalization, please refer to the appendix A.4. In Table 1, we present reconstruction metrics for some specific object categories as well as the overall performance across the entire dataset. It can be observed that our method consistently outperforms the baselines in both IoU and CD metrics. Complete results for all categories in each dataset are provided in A.4. We conducted a series of ablation studies to evaluate the impact of various components within our model. For comprehensive details on these experiments, please refer to A.4.

Table 1: Quantitative comparison of reconstruction performance between MeshCoder and baselines.

| Method | CD($\times 10^{-2}$)↓ | | | | | | IoU (%) ↑ | | | | | |
|---|---|---|---|---|---|---|---|---|---|---|---|---|
| | Lamp | Chair | Sofa | TableDining | Toilet | All | Lamp | Chair | Sofa | TableDining | Toilet | All |
| Shape2Prog | 25.44 | 1.30 | 2.14 | 1.03 | 7.51 | 6.01 | 16.96 | 49.68 | 65.29 | 71.26 | 51.14 | 45.03 |
| PLAD | 1.40 | 2.26 | 1.52 | 5.52 | 2.30 | 1.87 | 69.58 | 40.93 | 81.33 | 58.43 | 62.61 | 67.62 |
| **MeshCoder** | **0.004** | **0.060** | **0.027** | **0.024** | **0.022** | **0.063** | **86.23** | **81.87** | **93.81** | **88.14** | **89.10** | **86.75** |

### 4.4   Shape Editing

MeshCoder facilitates the transformation of 3D shapes into high-level, human-readable code representations, significantly enhancing the interpretability and editability of complex geometries. This capability enables intuitive and precise modifications through code-based interventions. Our shape editing encompasses two primary categories: geometric editing and topological editing. As illustrated in Figure 7, geometric editing can be performed by adjusting function calls or modifying specific parameters within the generated code. For instance, we can adjust the parameters of the code to convert a square tabletop into a larger circular one. Additionally, topological editing, which is illustrated in Figure 8 such as adjusting mesh resolution, can be achieved by modifying designated parameters within the code, allowing for control over the mesh's complexity and surface detail. This code-centric approach streamlines the process of modifying 3D models, making it more accessible and efficient for applications requiring iterative design and customization. Additionally, it empowers users to adjust the model resolution according to their desired balance between storage requirements and mesh quality. For additional results and details, please refer to A.5 in the appendix.

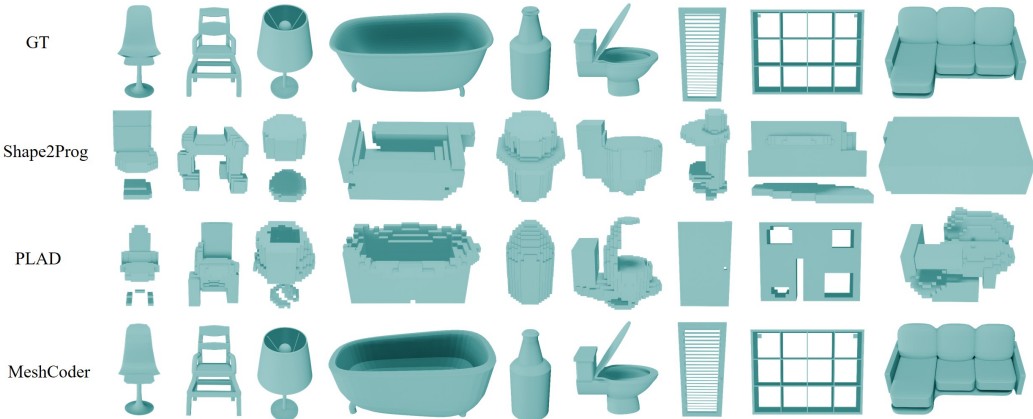

Figure 6: Qualitative comparison of reconstruction performance between MeshCoder and baselines. MeshCoder can accurately reconstruct objects with intricate parts and complex structures.

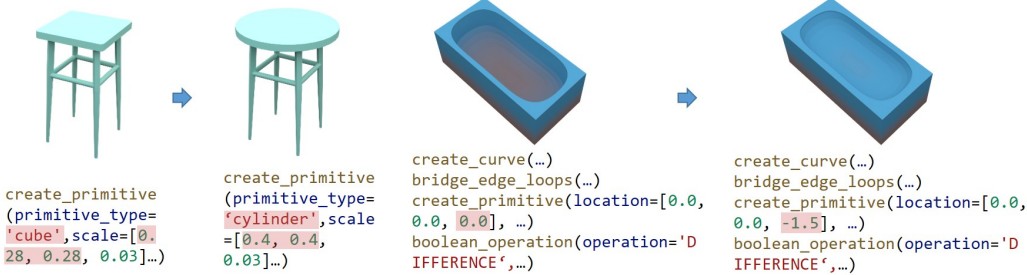

Figure 7: Parameter modification in the code conveniently to alter the geometric shape. Left: Change tabletop from square to circular. Right: Make the bathtub shallower.

## 4.5 Shape Undertanding

MeshCoder is capable of predicting object codes with rich semantic information. These codes effectively capture structural and geometric details, making them valuable for shape understanding. By inputting the predicted codes into GPT, we can assist it in comprehending object structures. We conduct experiments on shape understanding, with an example illustrated in Figure 9. Additional results and details are given in A.6 in the appendix.

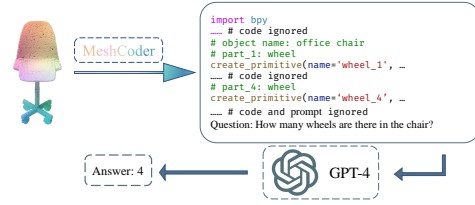

Figure 9: The pipeline of conducting experiments on shape understanding.

## 5  Limitations

Although our method achieves significant advancements in category diversity, geometric complexity, and reconstruction accuracy compared to existing approaches, it primarily targets human-made objects. The applicability of code-based representations to organic forms, such as animals and humans, remains underdeveloped. We reserve this as a direction for future research.

## 6  Conclusion

In this work, we present MeshCoder, a comprehensive framework that translates 3D point cloud data into editable Blender Python scripts, enabling detailed reconstruction and intuitive editing of complex 3D objects. By developing a robust set of Blender Python APIs, we facilitate the modeling of intricate geometries. Leveraging these APIs, we constructed a large-scale dataset pairing 3D objects with their corresponding code representations, decomposed into semantic parts. Subsequently, we

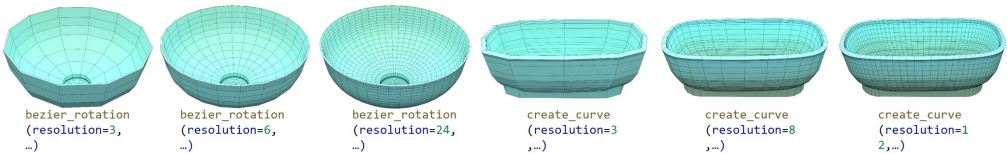

Figure 8: Mesh resolution adjustment by modifying the resolution parameters in the code. The figure depicts results with progressively increasing resolution from left to right.

trained a multimodal large language model (LLM) capable of generating executable Blender scripts from point cloud inputs. Our approach not only achieves superior performance in shape-to-code reconstruction tasks but also enhances the reasoning capabilities of LLMs in 3D shape understanding. By representing shapes as structured code, MeshCoder offers a flexible and powerful solution for programmatic 3D shape reconstruction and editing, paving the way for advanced applications in reverse engineering, design, and analysis.

# 7   Acknowledgement

This work is funded by the Shenzhen Science and Technology Project (Grants JCYJ20220818101001004 and KJZD20240903103210014), the National Key R&D Program of China (Grant 2022ZD0160201), Shanghai Artificial Intelligence Laboratory, and in part by the HKU Startup Fund.

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

# A Appendix of MeshCoder: LLM-Powered Structured Mesh Code Generation from Point Clouds

## A.1 Datasets

### A.1.1 The principles of Translation and Bridge Loop

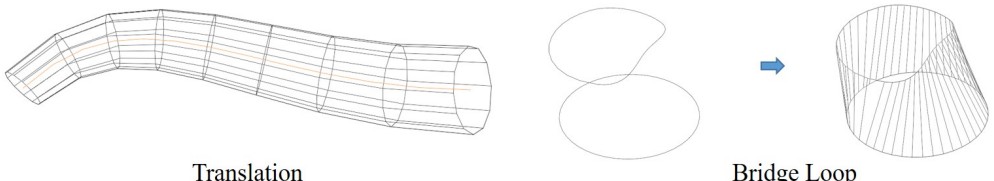

Translation                                    Bridge Loop

Figure 10: A schematic illustration of the principles of Translation and Bridge Loop. In the Translation module, the wireframe of the resulting mesh is shown as a cross-sectional circle is translated along a yellow trajectory. In the Bridge Loop module, the wireframe of the mesh is constructed by connecting the vertices of two 2D shapes.

As illustrated in the figure 10, in the Translation operation, a 2D cross-sectional shape (a circle in this example) and a 3D trajectory curve must first be defined. The Translation process generates a mesh by sweeping the 2D shape along the 3D trajectory. During this sweep, the cross-section remains perpendicular to the tangent direction of the trajectory at all times, and only uniform scaling (either enlargement or reduction) of the cross-section is permitted.

In contrast, the Bridge Loop operation begins with two predefined 2D shapes. By connecting the corresponding vertices of these two shapes, a mesh can be constructed. This method places no constraints on the types of 2D shapes used—meaning the two shapes can differ, such as a circle and a irregular closed shape in this example. Moreover, it imposes no restrictions on the relative orientations of the shapes. As a result, Bridge Loop overcomes the limitations of Translation, which requires the cross-section to align with the trajectory's tangent direction. This enables Bridge Loop to generate more complex geometries that Translation cannot produce.

### A.1.2 Part datasets

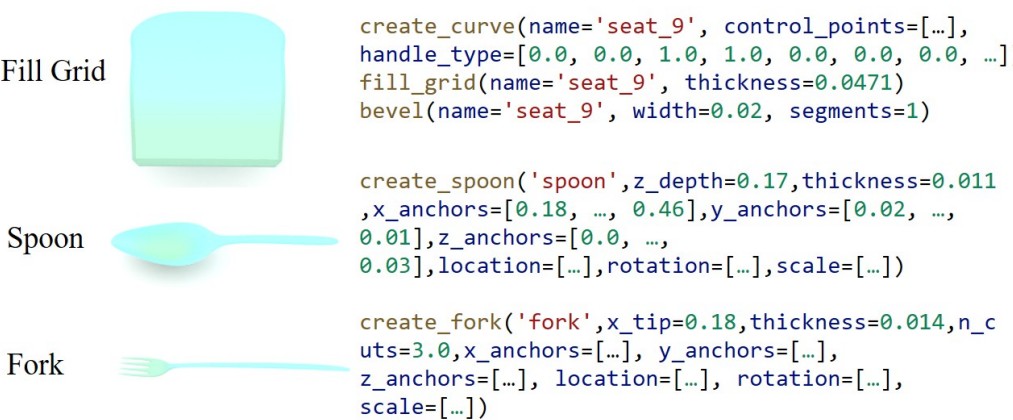

Figure 11: The Fill Grid type, Spoon type and Fork type in basic shape code library

For certain shapes that are difficult to represent using the method we defined in Section 3.1, we introduce three additional categories: the Fill Grid type, Spoon type and Fork type. As illustrated in the Figure 11. For the Fill Grid type, we first construct a closed 3D shape (as opposed to the 2D cross-sectional shape used in Translation), fill it to form a surface, and then extrude it along its

normal direction to generate the final mesh. For the Spoon and Fork type, we draw inspiration from the implementation in Infinigen Indoor [4] and design dedicated procedural functions tailored for their generation.

We present two core functions from our codebase: the complete implementation for creating primitives (Figure 22) and the complete implementation for creating curves (Figure 23). The full codebase can be found in the supplementary materials.

More examples of parts and their corresponding complete code implementations are provided in Figures 12, 13, 14, 15, and 16.

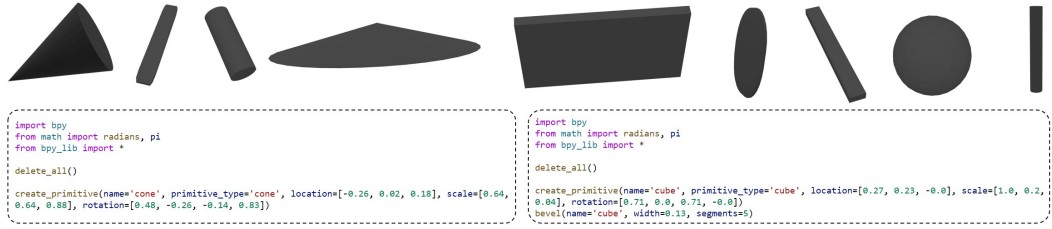

```
import bpy
from math import radians, pi
from bpy_lib import *

delete_all()

create_primitive(name='cone', primitive_type='cone', location=[-0.26, 0.02, 0.18], scale=[0.64,
0.64, 0.88], rotation=[0.48, -0.26, -0.14, 0.83])
```

```
import bpy
from math import radians, pi
from bpy_lib import *

delete_all()

create_primitive(name='cube', primitive_type='cube', location=[0.27, 0.23, -0.0], scale=[1.0, 0.2,
0.04], rotation=[0.71, 0.0, 0.71, -0.0])
bevel(name='cube', width=0.13, segments=5)
```

Figure 12: Examples of Primitive and complete code. And the code corresponds to the first two objects shown in the figure.

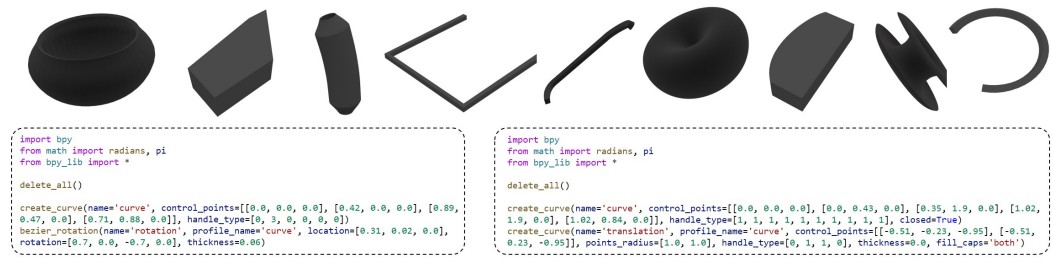

```
import bpy
from math import radians, pi
from bpy_lib import *

delete_all()

create_curve(name='curve', control_points=[[0.0, 0.0, 0.0], [0.42, 0.0, 0.0], [0.89,
0.47, 0.0], [0.71, 0.88, 0.0]], handle_type=[0, 3, 0, 0, 0])
bezier_rotation(name='rotation', profile_name='curve', location=[0.31, 0.02, 0.0],
rotation=[0.7, 0.0, -0.7, 0.0], thickness=0.06)
```

```
import bpy
from math import radians, pi
from bpy_lib import *

delete_all()

create_curve(name='curve', control_points=[[0.0, 0.0, 0.0], [0.0, 0.43, 0.0], [0.35, 1.9, 0.0], [1.02,
1.9, 0.0], [1.02, 0.84, 0.0]], handle_type=[1, 1, 1, 1, 1, 1, 1, 1], closed=True)
create_curve(name='translation', profile_name='curve', control_points=[[-0.51, -0.23, -0.95], [-0.51,
0.23, -0.95]], points_radius=[1.0, 1.0], handle_type=[0, 1, 1, 0], thickness=0.0, fill_caps='both')
```

Figure 13: Examples of Translation and complete code. And the code corresponds to the first two objects shown in the figure.

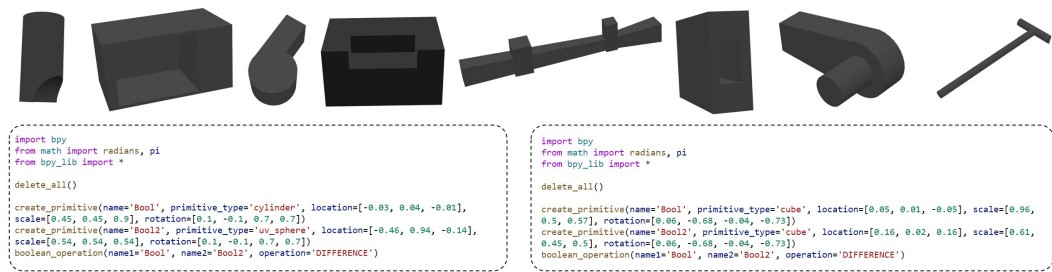

```
import bpy
from math import radians, pi
from bpy_lib import *

delete_all()

create_primitive(name='Bool', primitive_type='cylinder', location=[-0.03, 0.04, -0.01],
scale=[0.45, 0.45, 0.9], rotation=[0.1, -0.1, 0.7, 0.7])
create_primitive(name='Bool2', primitive_type='uv_sphere', location=[-0.46, 0.94, -0.14],
scale=[0.54, 0.54, 0.54], rotation=[0.1, -0.1, 0.7, 0.7])
boolean_operation(name1='Bool', name2='Bool2', operation='DIFFERENCE')
```

```
import bpy
from math import radians, pi
from bpy_lib import *

delete_all()

create_primitive(name='Bool', primitive_type='cube', location=[0.05, 0.01, -0.05], scale=[0.96,
0.5, 0.57], rotation=[0.06, -0.68, -0.04, -0.73])
create_primitive(name='Bool2', primitive_type='cube', location=[0.16, 0.02, 0.16], scale=[0.61,
0.45, 0.5], rotation=[0.06, -0.68, -0.04, -0.73])
boolean_operation(name1='Bool', name2='Bool2', operation='DIFFERENCE')
```

Figure 14: Examples of Boolean and complete code. And the code corresponds to the first two objects shown in the figure.

Taking the `Primitive` type as an example, we describe how to use functions from the basic shape code library to generate a synthetic part dataset. We begin by randomly selecting the type of primitive to generate (e.g., `cube`, `cylinder`, etc.). Next, for each axis, we independently uniform sample a value $x$ from the range $[-2, 2]$, and then set the corresponding scale as $10^x$. To determine the orientation of the shape, we uniformly sample a direction from a unit sphere and a roll angle from a uniform distribution. Once the orientation is fixed, we scale the shape uniformly along all three axes based on the size of its bounding box. Specifically, we ensure that the longest edge of the bounding box lies within the range $[1, 2]$. Finally, we assign the shape a random position within the 3D space such that the entire shape remains within the $[-1, 1]$ bounds. For other shape types beyond `Primitive`, we follow a similar approach by randomly assigning values to the relevant parameters.

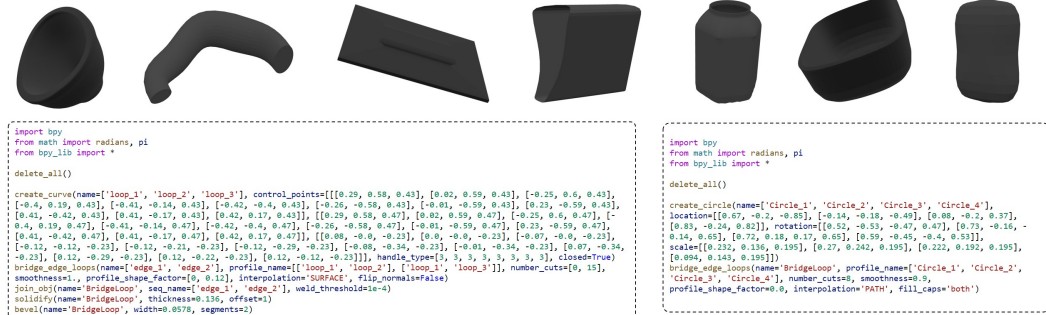

```
import bpy
from math import radians, pi
from bpy_lib import *

delete_all()

create_curve(name=['loop_1', 'loop_2', 'loop_3'], control_points=[[[0.29, 0.58, 0.43], [0.02, 0.59, 0.43], [-0.25, 0.6, 0.43],
[-0.4, 0.19, 0.43], [-0.41, -0.14, 0.43], [-0.42, -0.4, 0.43], [-0.26, -0.58, 0.43], [-0.01, -0.59, 0.43], [0.23, -0.59, 0.43],
[0.41, -0.42, 0.43], [0.41, -0.17, 0.43], [0.42, 0.17, 0.43]], [[0.29, 0.58, 0.47], [0.02, 0.59, 0.47], [-0.25, 0.6, 0.47], [-
0.4, 0.19, 0.47], [-0.41, -0.14, 0.47], [-0.42, -0.4, 0.47], [-0.26, -0.58, 0.47], [-0.01, -0.59, 0.47], [0.23, -0.59, 0.47],
[0.41, -0.42, 0.47], [0.41, -0.17, 0.47], [0.42, 0.17, 0.47]], [[0.08, -0.0, -0.23], [0.0, -0.0, -0.23], [-0.07, -0.0, -0.23],
[-0.12, -0.12, -0.23], [-0.12, -0.21, -0.23], [-0.12, -0.29, -0.23], [-0.08, -0.34, -0.23], [-0.01, -0.34, -0.23], [0.07, -0.34,
-0.23], [0.12, -0.29, -0.23], [0.12, -0.22, -0.23], [0.12, -0.12, -0.23]]], handle_type=[3, 3, 3, 3, 3, 3, 3, 3], closed=True)
bridge_edge_loops(name=['edge_1', 'edge_2'], profile_name=[['loop_1', 'loop_2'], ['loop_1', 'loop_3']], number_cuts=[0, 15],
smoothness=1, profile_shape_factor=[0, 0.12], interpolation='SURFACE', flip_normals=False)
join_obj(name='BridgeLoop', seq_name=['edge_1', 'edge_2'], weld_threshold=1e-4)
solidify(name='BridgeLoop', thickness=0.136, offset=1)
bevel(name='BridgeLoop', width=0.0578, segments=2)
```

```
import bpy
from math import radians, pi
from bpy_lib import *

delete_all()

create_circle(name=['Circle_1', 'Circle_2', 'Circle_3', 'Circle_4'],
location=[[0.67, -0.2, -0.85], [-0.14, -0.18, -0.49], [0.08, -0.2, 0.37],
[0.83, -0.24, 0.82]], rotation=[[0.52, -0.53, -0.47, 0.47], [0.73, -0.16, -
0.14, 0.65], [0.72, 0.18, 0.17, 0.65], [0.59, -0.45, -0.4, 0.53]],
scale=[[0.232, 0.136, 0.195], [0.27, 0.242, 0.195], [0.222, 0.192, 0.195],
[0.094, 0.143, 0.195]])
bridge_edge_loops(name='BridgeLoop', profile_name=['Circle_1', 'Circle_2',
'Circle_3', 'Circle_4'], number_cuts=8, smoothness=0.9,
profile_shape_factor=0.0, interpolation='PATH', fill_caps='both')
```

Figure 15: Examples of Bridge Loop and complete code. And the code corresponds to the first two objects shown in the figure.

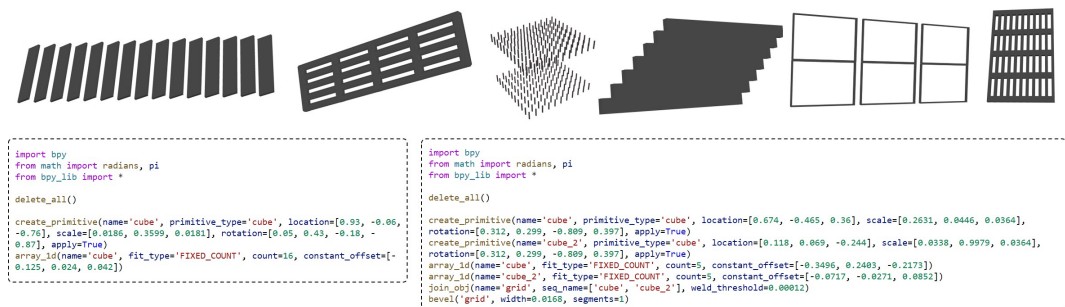

```
import bpy
from math import radians, pi
from bpy_lib import *

delete_all()

create_primitive(name='cube', primitive_type='cube', location=[0.93, -0.06,
-0.76], scale=[0.0186, 0.3599, 0.0181], rotation=[0.05, 0.43, -0.18, -
0.87], apply=True)
array_1d(name='cube', fit_type='FIXED_COUNT', count=16, constant_offset=[-
0.125, 0.024, 0.042])
```

```
import bpy
from math import radians, pi
from bpy_lib import *

delete_all()

create_primitive(name='cube', primitive_type='cube', location=[0.674, -0.465, 0.36], scale=[0.2631, 0.0446, 0.0364],
rotation=[0.312, 0.299, -0.809, 0.397], apply=True)
create_primitive(name='cube_2', primitive_type='cube', location=[0.118, 0.069, -0.244], scale=[0.0338, 0.9979, 0.0364],
rotation=[0.312, 0.299, -0.809, 0.397], apply=True)
array_1d(name='cube', fit_type='FIXED_COUNT', count=5, constant_offset=[-0.3496, 0.2403, -0.2173])
array_1d(name='cube_2', fit_type='FIXED_COUNT', count=5, constant_offset=[-0.0717, -0.0271, 0.0852])
join_obj(name='grid', seq_name=['cube', 'cube_2'], weld_threshold=0.00012)
bevel('grid', width=0.0168, segments=1)
```

Figure 16: Examples of Array and complete code. And the code corresponds to the first two objects shown in the figure.

### A.1.3 Object datasets

For assembling part codes into a complete program, we provide a full example containing the complete code, as shown in Figure 17. Regarding the ordering strategy used when assembling parts into a complete object, we adopt a consistent spatial heuristic to determine part sequence. Specifically, parts are arranged from bottom to top, left to right, and front to back. To implement this, we divide the 3D space into a $32 \times 32 \times 32$ grid and assign each part a characteristic grid cell that serves as the basis for sorting. The characteristic grid cell of a part is defined as follows: among all grid cells that the part occupies, we first select the one with the smallest $z$-coordinate. If multiple candidates share the same $z$-value, we choose the one with the smallest $x$-coordinate. If a tie still exists, we select the one with the smallest $y$-coordinate. Parts are then sorted based on the lexicographic order of these characteristic grid cells, which determines their final sequence within the object.

It is important to note that for each object, the prerequisite for successfully constructing its corresponding code lies in the ability of our part-to-code inference model to accurately infer all of its individual parts. We consider a part to be successfully inferred if the Chamfer Distance (CD) between the predicted point cloud and the ground truth is below $5 \times 10^{-3}$. Therefore, when constructing the object-code pairs dataset, we only include objects for which **all** constituent parts meet this criterion. Objects with any part failing to meet this standard are discarded. As a result, the number of successfully constructed object-code pairs is smaller than the total number of objects in the original Infinigen dataset. In fact, the original Infinigen dataset we use contains 1.57 million object instances, from which we successfully construct 1 million shape-code pairs. For training and evaluation, we split the full Infinigen dataset into 70% for training, 15% for testing, and 15% for validation. Accordingly, MESHCODER is trained only on the subset of the shape-code pairs that fall within the training portion of the Infinigen dataset. In contrast, the baseline models are trained on the full set of objects in the training split of the original Infinigen dataset. Importantly, all evaluation results for our method and the baselines are reported on the same test set, i.e., the testing split of the complete Infinigen dataset.

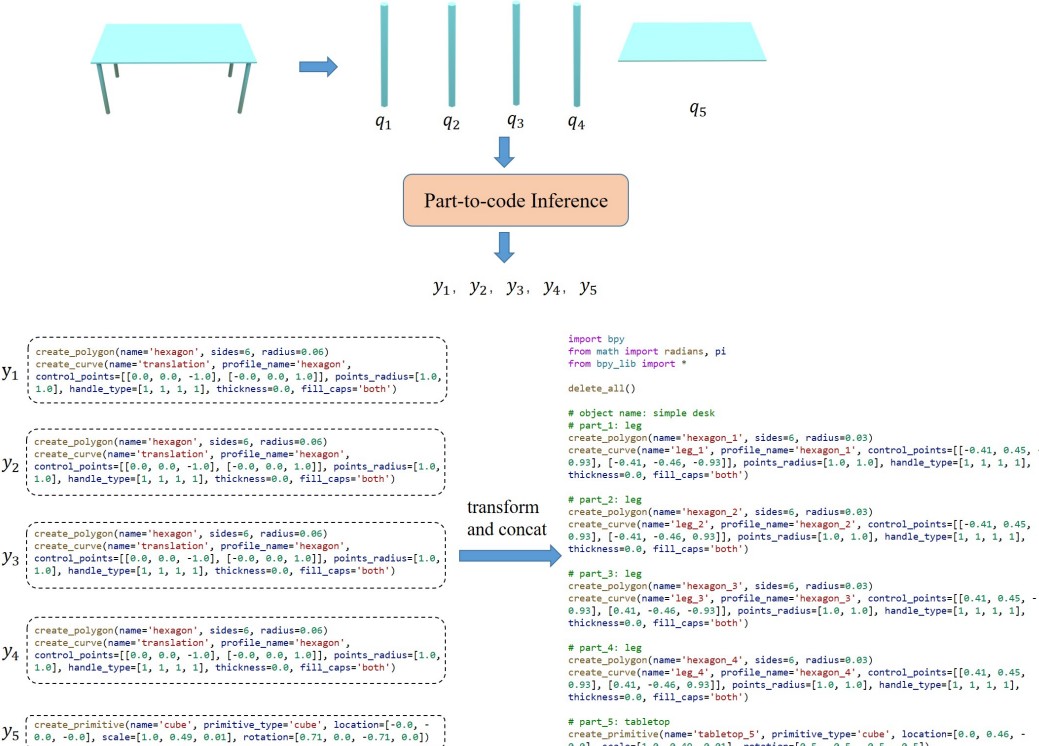

Figure 17: A complete code example of converting part codes into a full object program.

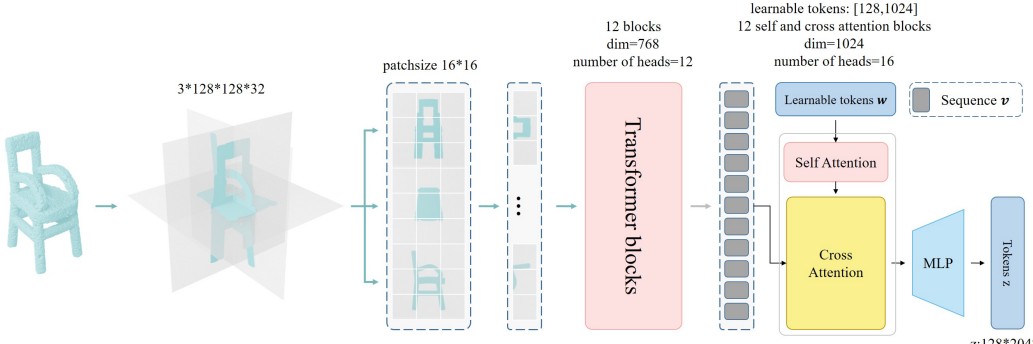

Figure 18: Detailed configuration of the shape tokenizer.

## A.2 Model architecture

We explain the detailed structure of the shape tokenizer. As illustrated in the Figure 18, we first project the input point cloud of shape $\mathbb{R}^{n \times 3}$ onto three orthogonal planes to obtain tri-plane features with shape $\mathbb{R}^{3 \times 256 \times 256 \times 32}$. With a patch size set to $16 \times 16$, these tri-plane features are encoded into tokens and fed into Transformer blocks, where the resulting representation is mapped to $v$ and used as the key and value $(K, V)$ inputs. Meanwhile, a set of learnable tokens with shape $\mathbb{R}^{128 \times 1024}$ are used as queries in a self and cross attention module. After passing through 12 layers of self and cross attention, we obtain output tokens of shape $\mathbb{R}^{128 \times 1024}$, which are then projected to the final representation of shape $\mathbb{R}^{128 \times 2048}$ via an MLP.

## A.3 More training details

For the part-to-code reconstruction model, we adopt the AdamW optimizer and train it for 20 epochs on 64 NVIDIA A100 GPUs for about a week with a batch size of 512, and a learning rate of $10^{-4}$. We evaluate the model at every epoch and select the checkpoint with the lowest $L_2$ Chamfer Distance (CD) loss. Then we initialize the weights of the object-to-code reconstruction model with the weights of the trained part-to-code reconstruction model, and train the model on Infinigen Indoor dataset for 10 epochs, with a batch size of 256, and a learning rate of $10^{-4}$. It is trained on 64 NVIDIA A100 GPUs for about 2 days. The checkpoint with the lowest CD loss is selected.

To further enhance the robustness and generalization ability of the object-to-code inference model, we apply data augmentation techniques. Specifically, we perform random rotation and scaling on the objects. Additionally, during training, we randomly sample the number of points in each point cloud within the range of 4096 to 16384, and add Gaussian noise to further perturb the input. MeshCoder is trained and evaluated on a unified dataset that aggregates all object categories.

## A.4 Complete experiment result of Shape Reconstruction

For **MeshCoder**, during inference, each object is represented by a point cloud containing 16,384 points. Given the input point cloud, the object-to-code inference model is able to predict the corresponding Blender Python script code. The resulting code is then executed to generate a corresponding mesh. We uniformly sample 100,000 points from the generated mesh and compute the Chamfer Distance (CD) to the input point cloud using the $L_2$ norm.

Given two point sets $P$ and $Q$, each of size 100,000, the $L_2$ Chamfer Distance is defined as:

$$\text{CD}(P, Q) = \frac{1}{|P|} \sum_{x \in P} \min_{y \in Q} \|x - y\|_2^2 + \frac{1}{|Q|} \sum_{y \in Q} \min_{x \in P} \|y - x\|_2^2.$$

To evaluate IoU, we voxelize both the ground-truth mesh and the predicted mesh into grids of resolution $32^3$, and compute the voxel-based Intersection-over-Union (IoU) as:

$$\text{IoU} = \frac{|\mathcal{V}_{\text{pred}} \cap \mathcal{V}_{\text{gt}}|}{|\mathcal{V}_{\text{pred}} \cup \mathcal{V}_{\text{gt}}|},$$

where $\mathcal{V}_{\text{pred}}$ and $\mathcal{V}_{\text{gt}}$ denote the sets of occupied voxels in the predicted and ground-truth voxel grids, respectively.

For **baseline methods**, which take voxel grids as input and output voxel grids, we first voxelize the ground-truth mesh into a $32^3$ grid and feed it into the baseline models. The predicted voxel grid is then compared to the input voxelized ground truth to compute IoU. Additionally, we extract a mesh from the predicted voxel grid using the Marching Cubes algorithm and uniformly sample 100,000 points from the resulting mesh surface. These sampled points, along with the ground-truth point cloud, are then both uniformly scaled to fit within the $[-1, 1]^3$ volume. Finally, the Chamfer Distance is computed between the two normalized point clouds using the $L_2$ norm.

It's noticed that for each object category, we independently train the baseline models, according to their official code, resulting in category-specific checkpoints. These models are then evaluated on the corresponding test sets for each category.

The quantitative comparison of reconstruction metrics between MeshCoder and baseline methods across all object categories is summarized in Table 2 and Table 3. Some additional examples of object reconstruction results and their complete code can be referred to Figure 24, 25, 26.

In addition to evaluating our object-to-code inference model, we also perform a quantitative assessment of our part-to-code inference model. Specifically, for each category described in Section 3.1, we construct a test set consisting of 10,000 samples. We evaluate the model's performance using the CD and voxel IoU metrics on these test sets. The results, shown in Table 4, demonstrate strong performance across all categories, with low CD values and high IoU scores, indicating that our part-to-code inference model is highly effective in generating accurate code representations for individual parts.

Table 2: Comparison of reconstruction metrics across all categories. Chamfer Distance (CD) and IoU is shown in percentage (%).

| Category | L2 CD($\times 10^{-2}$) | | | Voxel IoU (%) | | |
| --- | --- | --- | --- | --- | --- | --- |
| | MeshCoder | PLAD | Shape2prog | MeshCoder | PLAD | Shape2prog |
| ArmChair | 0.04 | 2.31 | 4.44 | 94.33 | 78.79 | 62.74 |
| BarChair | 0.03 | 2.23 | 2.55 | 88.73 | 74.96 | 58.23 |
| Bathtub | 0.09 | 1.22 | 2.45 | 78.70 | 74.50 | 42.94 |
| BeverageFridge | 0.22 | 1.12 | 12.63 | 88.03 | 82.13 | 39.13 |
| Bottle | 0.01 | 1.08 | 6.34 | 88.65 | 65.58 | 40.24 |
| Bowl | 0.02 | 1.43 | 6.29 | 89.93 | 60.02 | 25.60 |
| CeilingClassicLamp | 0.02 | 1.98 | 3.94 | 96.13 | 76.01 | 59.07 |
| CeilingLight | 0.03 | 3.46 | 1.32 | 65.83 | 40.61 | 44.97 |
| CellShelf | 0.01 | 1.93 | 9.40 | 94.67 | 59.02 | 22.30 |
| Chair | 0.06 | 2.26 | 1.30 | 81.87 | 40.93 | 49.68 |
| Chopsticks | 0.03 | 1.38 | 21.06 | 82.24 | 55.68 | 11.25 |
| Cup | 0.06 | 1.40 | 7.35 | 85.96 | 62.03 | 29.47 |
| DeskLamp | 0.02 | 1.76 | 8.77 | 80.28 | 64.31 | 25.35 |
| Dishwasher | 0.13 | 1.44 | 3.01 | 88.37 | 84.44 | 46.69 |
| FloorLamp | 0.00 | 2.13 | 22.97 | 85.96 | 66.89 | 17.16 |
| Fork | 0.14 | 0.34 | 8.40 | 58.86 | 89.28 | 11.03 |
| Hardware | 0.01 | 0.62 | 8.45 | 89.87 | 83.96 | 23.56 |
| Jar | 0.03 | 0.76 | 1.39 | 79.12 | 69.67 | 41.51 |
| Lamp | 0.00 | 1.40 | 25.44 | 86.23 | 69.58 | 16.96 |
| LargeShelf | 0.02 | 0.82 | 5.15 | 88.08 | 60.70 | 16.81 |
| Lid | 0.05 | 1.83 | 2.39 | 73.22 | 63.47 | 50.11 |
| LiteDoor | 0.03 | 1.36 | 5.75 | 94.75 | 36.91 | 18.71 |
| LouverDoor | 0.07 | 1.40 | 16.17 | 89.46 | 37.43 | 20.94 |
| Microwave | 0.07 | 1.44 | 11.04 | 91.72 | 55.65 | 49.38 |
| OfficeChair | 0.03 | 1.44 | 2.63 | 78.41 | 55.65 | 46.91 |
| PanelDoor | 0.04 | 1.31 | 6.50 | 94.60 | 37.18 | 20.94 |
| Plate | 0.04 | 0.96 | 1.07 | 72.70 | 70.72 | 60.05 |
| SidetableDesk | 0.01 | 0.67 | 4.50 | 93.23 | 91.75 | 35.75 |
| SimpleBookcase | 0.03 | 1.78 | 2.89 | 92.14 | 65.14 | 33.79 |
| SimpleDesk | 0.01 | 2.12 | 25.39 | 88.68 | 93.80 | 45.79 |
| Sofa | 0.03 | 1.52 | 2.14 | 93.81 | 81.33 | 65.29 |
| Spoon | 0.67 | 0.37 | 4.09 | 74.00 | 87.04 | 18.92 |
| TableCocktail | 0.02 | 2.59 | 5.93 | 88.47 | 60.49 | 25.19 |
| TableDining | 0.02 | 5.52 | 1.03 | 88.14 | 58.43 | 71.26 |
| Toilet | 0.02 | 2.30 | 7.51 | 89.10 | 62.61 | 51.14 |
| TriangleShelf | 0.01 | 2.30 | 12.61 | 88.75 | 62.61 | 30.59 |
| TV | 0.04 | 1.53 | 3.41 | 87.80 | 72.69 | 34.14 |
| TVStand | 0.01 | 0.78 | 13.50 | 91.26 | 73.78 | 22.57 |
| Vase | 0.30 | 0.73 | 19.10 | 72.26 | 89.95 | 60.94 |
| Window | 0.14 | 0.59 | 3.73 | 87.36 | 84.21 | 64.64 |
| Wineglass | 0.06 | 0.98 | 6.83 | 88.36 | 73.96 | 28.56 |
| **All (Avg.)** | **0.06** | **1.87** | **6.00** | **86.75** | **67.62** | **45.03** |

### A.4.1  Ablation Study

We conduct four ablation studies to evaluate the impact of key design choices in our framework.

**Triplane Resolution and the Number of Learnable Tokens.**  The first ablation investigates the effect of varying the resolution of the triplane representation and the number of learnable tokens. As shown in Table 5, we observe that increasing both the triplane resolution and the number of learnable tokens consistently improves the performance of the object-to-code inference model. This suggests that a finer-grained spatial encoding and a richer set of token representations enable the model to better capture the underlying 3D structure of objects.

Table 3: Comparison of standard deviation of reconstruction metrics across all categories.

| Category | CD | | | IoU | | |
|---|---|---|---|---|---|---|
| | MeshCoder | PLAD | Shape2prog | MeshCoder | PLAD | Shape2prog |
| ArmChair | $1.51 \times 10^{-3}$ | $9.35 \times 10^{-3}$ | $1.51 \times 10^{-2}$ | $4.62 \times 10^{-2}$ | $6.48 \times 10^{-2}$ | $5.28 \times 10^{-2}$ |
| BarChair | $1.82 \times 10^{-4}$ | $1.18 \times 10^{-2}$ | $9.90 \times 10^{-3}$ | $8.19 \times 10^{-2}$ | $1.00 \times 10^{-1}$ | $8.30 \times 10^{-2}$ |
| Bathtub | $6.93 \times 10^{-4}$ | $1.02 \times 10^{-2}$ | $6.70 \times 10^{-3}$ | $1.31 \times 10^{-1}$ | $1.91 \times 10^{-1}$ | $8.57 \times 10^{-2}$ |
| BeverageFridge | $4.84 \times 10^{-3}$ | $2.81 \times 10^{-3}$ | $3.44 \times 10^{-2}$ | $1.13 \times 10^{-1}$ | $6.23 \times 10^{-2}$ | $6.64 \times 10^{-2}$ |
| Bottle | $7.56 \times 10^{-5}$ | $6.80 \times 10^{-3}$ | $6.00 \times 10^{-2}$ | $1.13 \times 10^{-1}$ | $7.05 \times 10^{-2}$ | $6.90 \times 10^{-2}$ |
| Bowl | $4.83 \times 10^{-5}$ | $5.13 \times 10^{-3}$ | $8.78 \times 10^{-3}$ | $8.11 \times 10^{-2}$ | $6.24 \times 10^{-2}$ | $2.35 \times 10^{-2}$ |
| CeilingClassicLamp | $7.33 \times 10^{-7}$ | $7.66 \times 10^{-4}$ | $9.86 \times 10^{-4}$ | $3.39 \times 10^{-5}$ | $2.96 \times 10^{-3}$ | $3.82 \times 10^{-2}$ |
| CeilingLight | $1.79 \times 10^{-6}$ | $3.90 \times 10^{-3}$ | $4.44 \times 10^{-3}$ | $3.10 \times 10^{-1}$ | $5.08 \times 10^{-2}$ | $6.93 \times 10^{-2}$ |
| CellShelf | $3.37 \times 10^{-5}$ | $1.94 \times 10^{-2}$ | $6.95 \times 10^{-2}$ | $9.65 \times 10^{-2}$ | $1.34 \times 10^{-1}$ | $9.45 \times 10^{-2}$ |
| Lamp | $2.20 \times 10^{-5}$ | $9.05 \times 10^{-3}$ | $2.74 \times 10^{-1}$ | $1.56 \times 10^{-1}$ | $6.87 \times 10^{-2}$ | $1.18 \times 10^{-1}$ |
| Chair | $1.09 \times 10^{-3}$ | $1.04 \times 10^{-2}$ | $4.52 \times 10^{-3}$ | $1.05 \times 10^{-1}$ | $9.17 \times 10^{-2}$ | $6.72 \times 10^{-2}$ |
| Chopsticks | $3.64 \times 10^{-3}$ | $1.31 \times 10^{-2}$ | $1.85 \times 10^{-1}$ | $1.87 \times 10^{-1}$ | $1.00 \times 10^{-1}$ | $1.01 \times 10^{-1}$ |
| Cup | $1.59 \times 10^{-3}$ | $5.79 \times 10^{-3}$ | $3.60 \times 10^{-2}$ | $9.84 \times 10^{-2}$ | $6.80 \times 10^{-2}$ | $6.98 \times 10^{-2}$ |
| DeskLamp | $7.62 \times 10^{-4}$ | $8.60 \times 10^{-3}$ | $4.55 \times 10^{-2}$ | $1.30 \times 10^{-1}$ | $7.21 \times 10^{-2}$ | $6.01 \times 10^{-2}$ |
| Dishwasher | $9.66 \times 10^{-3}$ | $2.69 \times 10^{-3}$ | $2.39 \times 10^{-2}$ | $1.27 \times 10^{-1}$ | $4.74 \times 10^{-2}$ | $8.82 \times 10^{-2}$ |
| FloorLamp | $1.23 \times 10^{-4}$ | $2.09 \times 10^{-2}$ | $2.54 \times 10^{-1}$ | $1.68 \times 10^{-1}$ | $4.92 \times 10^{-2}$ | $1.12 \times 10^{-1}$ |
| Fork | $8.81 \times 10^{-3}$ | $2.14 \times 10^{-3}$ | $8.57 \times 10^{-2}$ | $2.14 \times 10^{-1}$ | $1.25 \times 10^{-1}$ | $6.55 \times 10^{-2}$ |
| Hardware | $2.20 \times 10^{-4}$ | $3.07 \times 10^{-3}$ | $4.48 \times 10^{-2}$ | $1.21 \times 10^{-1}$ | $1.02 \times 10^{-1}$ | $1.34 \times 10^{-1}$ |
| Jar | $1.40 \times 10^{-4}$ | $2.44 \times 10^{-3}$ | $6.11 \times 10^{-3}$ | $1.44 \times 10^{-1}$ | $6.31 \times 10^{-2}$ | $8.98 \times 10^{-2}$ |
| LargeShelf | $1.79 \times 10^{-4}$ | $4.65 \times 10^{-3}$ | $5.12 \times 10^{-2}$ | $1.53 \times 10^{-1}$ | $8.67 \times 10^{-2}$ | $7.09 \times 10^{-2}$ |
| Lid | $8.89 \times 10^{-4}$ | $1.09 \times 10^{-2}$ | $1.95 \times 10^{-2}$ | $1.55 \times 10^{-1}$ | $1.22 \times 10^{-1}$ | $1.23 \times 10^{-1}$ |
| LiteDoor | $5.79 \times 10^{-3}$ | $4.39 \times 10^{-3}$ | $2.88 \times 10^{-2}$ | $1.44 \times 10^{-1}$ | $6.32 \times 10^{-2}$ | $9.69 \times 10^{-2}$ |
| LouverDoor | $4.67 \times 10^{-3}$ | $4.84 \times 10^{-3}$ | $9.23 \times 10^{-2}$ | $1.65 \times 10^{-1}$ | $6.82 \times 10^{-2}$ | $1.53 \times 10^{-1}$ |
| Microwave | $3.92 \times 10^{-3}$ | $2.43 \times 10^{-2}$ | $3.15 \times 10^{-2}$ | $7.26 \times 10^{-2}$ | $1.34 \times 10^{-1}$ | $1.65 \times 10^{-1}$ |
| OfficeChair | $1.72 \times 10^{-4}$ | $7.35 \times 10^{-3}$ | $2.95 \times 10^{-2}$ | $8.97 \times 10^{-2}$ | $1.06 \times 10^{-1}$ | $1.05 \times 10^{-1}$ |
| PanelDoor | $9.05 \times 10^{-3}$ | $4.79 \times 10^{-3}$ | $3.74 \times 10^{-2}$ | $1.50 \times 10^{-1}$ | $7.17 \times 10^{-2}$ | $1.09 \times 10^{-1}$ |
| Plate | $1.73 \times 10^{-4}$ | $6.40 \times 10^{-3}$ | $5.78 \times 10^{-3}$ | $1.70 \times 10^{-1}$ | $1.29 \times 10^{-1}$ | $1.74 \times 10^{-1}$ |
| SidetableDesk | $5.11 \times 10^{-5}$ | $3.52 \times 10^{-3}$ | $5.37 \times 10^{-2}$ | $9.64 \times 10^{-2}$ | $5.83 \times 10^{-2}$ | $1.23 \times 10^{-1}$ |
| SimpleBookcase | $3.66 \times 10^{-3}$ | $6.54 \times 10^{-3}$ | $7.01 \times 10^{-2}$ | $1.08 \times 10^{-1}$ | $9.62 \times 10^{-2}$ | $6.06 \times 10^{-2}$ |
| SimpleDesk | $4.29 \times 10^{-5}$ | $9.90 \times 10^{-2}$ | $1.72 \times 10^{-1}$ | $1.68 \times 10^{-1}$ | $6.43 \times 10^{-2}$ | $8.00 \times 10^{-2}$ |
| Sofa | $1.35 \times 10^{-3}$ | $5.78 \times 10^{-3}$ | $6.99 \times 10^{-3}$ | $6.61 \times 10^{-2}$ | $7.32 \times 10^{-2}$ | $6.37 \times 10^{-2}$ |
| Spoon | $5.64 \times 10^{-2}$ | $1.63 \times 10^{-3}$ | $4.59 \times 10^{-2}$ | $2.26 \times 10^{-1}$ | $8.26 \times 10^{-2}$ | $9.81 \times 10^{-2}$ |
| TableCocktail | $2.03 \times 10^{-4}$ | $2.85 \times 10^{-2}$ | $3.10 \times 10^{-2}$ | $1.09 \times 10^{-1}$ | $2.11 \times 10^{-1}$ | $8.68 \times 10^{-2}$ |
| TableDining | $3.31 \times 10^{-3}$ | $7.18 \times 10^{-2}$ | $6.16 \times 10^{-3}$ | $1.55 \times 10^{-1}$ | $1.64 \times 10^{-1}$ | $9.41 \times 10^{-2}$ |
| Toilet | $1.09 \times 10^{-4}$ | $8.44 \times 10^{-3}$ | $1.99 \times 10^{-2}$ | $4.22 \times 10^{-2}$ | $5.24 \times 10^{-2}$ | $5.41 \times 10^{-2}$ |
| TriangleShelf | $3.60 \times 10^{-5}$ | $9.30 \times 10^{-3}$ | $9.47 \times 10^{-2}$ | $1.03 \times 10^{-1}$ | $6.63 \times 10^{-2}$ | $1.08 \times 10^{-1}$ |
| TV | $6.25 \times 10^{-4}$ | $1.74 \times 10^{-3}$ | $1.02 \times 10^{-2}$ | $1.66 \times 10^{-1}$ | $2.84 \times 10^{-2}$ | $6.68 \times 10^{-2}$ |
| TVStand | $2.59 \times 10^{-5}$ | $1.45 \times 10^{-3}$ | $5.63 \times 10^{-2}$ | $1.31 \times 10^{-1}$ | $9.48 \times 10^{-2}$ | $5.92 \times 10^{-2}$ |
| Vase | $9.97 \times 10^{-3}$ | $3.44 \times 10^{-3}$ | $1.05 \times 10^{-1}$ | $2.68 \times 10^{-1}$ | $2.48 \times 10^{-2}$ | $4.00 \times 10^{-2}$ |
| Window | $9.57 \times 10^{-3}$ | $3.51 \times 10^{-3}$ | $6.61 \times 10^{-2}$ | $1.81 \times 10^{-1}$ | $1.14 \times 10^{-1}$ | $1.88 \times 10^{-1}$ |
| Wineglass | $1.40 \times 10^{-2}$ | $3.12 \times 10^{-3}$ | $3.86 \times 10^{-2}$ | $1.04 \times 10^{-1}$ | $5.39 \times 10^{-2}$ | $6.99 \times 10^{-2}$ |
| **All (Std.)** | $\mathbf{2.92 \times 10^{-3}}$ | $\mathbf{2.49 \times 10^{-2}}$ | $\mathbf{7.23 \times 10^{-2}}$ | $\mathbf{1.25 \times 10^{-1}}$ | $\mathbf{1.94 \times 10^{-1}}$ | $\mathbf{1.92 \times 10^{-1}}$ |

**Initialization from Part-to-Code Checkpoint.** The second ablation study evaluates whether initializing the object-to-code model with the pre-trained checkpoint of the part-to-code inference model yields performance improvements. Table 6 demonstrates that such initialization leads to noticeably better results. This improvement may be attributed to the part-to-code model's ability to learn robust 3D geometric representations and syntactic grammar structures from the diverse part-level dataset. These learned features likely provide transferable knowledge that facilitates generalization during the object-level inference process, thereby improving the effectiveness of the model.

**Using Learnable Tokens as Queries.** The third ablation study explores the role of learnable tokens when used as queries for the LLM. In our default setup, learnable tokens are used as input queries to the LLM, whereas in the alternative setup, the triplane-encoded features are passed through an MLP and directly fed into the LLM. As reported in Table 7, the learnable-token-as-query strategy achieves superior performance. We hypothesize that this advantage arises for two reasons: (1) the learnable tokens are capable of aggregating global information across the entire input, unlike the direct feature approach where each patch predominantly captures localized information, and (2) the learnable tokens can adaptively organize the input representation in a layout that is more aligned with the LLM's internal understanding and processing patterns.

Table 4: Quantitative evaluation of the *part-to-code* inference model across different part categories. CD is reported in $10^{-2}$, and IoU is reported in percentage.

| Category | CD ($\times 10^{-2}$) | IoU (%) |
|---|---|---|
| Primitive | 0.18 | 94.81 |
| Boolean | 0.03 | 96.13 |
| Array | 0.70 | 78.90 |
| Bridge Loop | 0.14 | 89.16 |
| Translation | 0.17 | 83.45 |

Table 5: Ablation study on triplane resolution and the number of learnable tokens in MeshCoder. We report L2 Chamfer Distance ($\times 10^{-4}$) and IoU (%).

| Triplane Resolution | Token Number | L2 CD ($\times 10^{-4}$) | IoU (%) |
|---|---|---|---|
| 256 | 128 | **6.32** | **86.75** |
| 256 | 64 | 7.72 | 85.11 |
| 256 | 32 | 7.05 | 85.02 |
| 128 | 128 | 6.55 | 86.62 |
| 64 | 128 | 6.51 | 86.05 |
| 32 | 128 | 6.88 | 84.56 |

**Ablate triplane.** To investigate the effectiveness of different point cloud encoding methods, we conducted an ablation study focusing on the triplane encoding strategy. Specifically, we compared it with the point cloud encoding approach from Hunyuan3D [37]: each input point is treated as an independent token, and its information is directly transferred to learnable tokens via cross-attention, bypassing the triplane intermediate representation. The experimental results are presented in the table, which demonstrate that the tri-plane encoding method achieves superior performance on the target task.

### A.5 Complete experiment result of Shape Editing

We additionally present two examples of shape editing along with their complete code implementations. In Figure 27, we modify the thickness of the chair legs and armrests by adjusting the `scale` parameter. In Figure 28, we change the mesh resolution of a plate by modifying the `resolution` parameter.

### A.6 Complete experiment result of Shape Understanding

When presented with a 3D point cloud of an object as input, MeshCoder can infer the corresponding code for the object. Upon execution of this code in Blender, the geometry of the object can be obtained. Notably, the comments within the code encompass a variety of semantically rich cues, such as the object's identity and the specifics of each component. The primary aim of this experiment is to highlight that our model can assist existing large language models, like GPT-4, in understanding the structure of 3D objects. We provide the inferred code to GPT-4 with the implementation details and functional descriptions of the functions used in the reconstruction code. Then, we input the object's reconstruction code itself and then inquire about the geometry or structure of the object, as showed in Figure 19, Figure 20 and Figure 21. GPT - 4 is able to generate relevant responses based on the code. More importantly, we found that this code-based representation has unique advantages over visual inputs like multi-view images. For instance, code enables the LLM to understand complex internal structures and provide precise dimensional measurements, which are very difficult to ascertain from images alone. Our further experiments demonstrate that code and multi-view images are complementary; providing both to the LLM leads to a higher accuracy than using either input in isolation. This demonstrates that our model possesses capabilities in understanding the geometry and structure of 3D objects and can aid large - scale models such as GPT in addressing such questions. However, our model does have limitations. Currently, the code inferred by our model solely contains geometric information of the object and does not include the color or texture information that images provide. As a result, it is unable to answer questions pertaining to color.

Table 6: Ablation study on whether to initialize object-to-code model from the part-to-code checkpoint.

| Initialization Strategy | L2 CD ($\times 10^{-4}$) | IoU (%) |
|---|---|---|
| From Scratch | 8.62 | 85.16 |
| From Part-to-Code Checkpoint | **6.32** | **86.75** |

Table 7: Ablation study on whether to use learnable tokens as queries in the transformer.

| Query Type | L2 CD ($\times 10^{-4}$) | IoU (%) |
|---|---|---|
| MLP Projection Only | 9.88 | 84.12 |
| Learnable Tokens (Ours) | **6.32** | **86.75** |

## A.7 More comparative experiments

To demonstrate our model's ability to generalize beyond the Infinigen dataset, we have conducted reconstruction experiments using non-Infinigen data. Specifically, we leveraged Trellis[38], a currently popular generative model that takes single images as input and outputs corresponding meshes, to create our experimental data. We collected images of three categories (chairs, shelves, and bottles) from online sources, with 10 images collected for each category. These images were input into Trellis to generate meshes, from which we sampled point clouds to serve as input for MeshCoder. To further demonstrate that our model does not merely memorize the training data, we selected examples from our training dataset that have the smallest CD loss with the point clouds generated by Trellis. All metric results are presented in Table 9.

This experiment shows that our model maintains satisfactory performance on non-Infinigen data, demonstrating its generalization capability. Additionally, we argue that the model does not merely memorize Infinigen-specific structures. If it did, the metrics of our model's predictions are unlikely to outperform those of the dataset matches, which represent the closest possible examples from the training set. However, we found that the reconstruction results are suboptimal when attempting to reconstruct objects from categories not present in the training dataset. To further strengthen generalization in future work, we plan to expand to more extensive and diverse datasets such as Objaverse[39], and incorporate reinforcement learning techniques (DPO[40], GRPO[41]) into our training pipeline, like DeepMesh[42], to boost the model's ability to handle broader data distributions.

Moreover, to investigate how varying point cloud densities affect reconstruction quality, we designed experiments with three different point cloud quantities, where the number of input points for our trained MeshCoder was set to 4096, 8192, and 16384 respectively to observe its reconstruction metrics across four representative categories: bottles, cell shelves, chairs, and sofas. The metric results are shown in Table 10. Notably, we observe a clear trend: as the number of input points increases, the reconstruction quality consistently improves.

**Table 8: Ablation study on triplane**

| Encoding approach Strategy | L2 CD ($\times 10^{-4}$) | IoU (%) |
|---|---|---|
| Treat point-cloud as token | 8.96 | 85.17 |
| Triplane | **6.32** | **86.75** |

**Table 9: Metrics of different methods on non-Infinigen data.**

| Method | Chairs | | Shelves | | Bottles | |
|---|---|---|---|---|---|---|
| | CD Loss ($\downarrow$) | IoU ($\uparrow$) | CD Loss ($\downarrow$) | IoU ($\uparrow$) | CD Loss ($\downarrow$) | IoU ($\uparrow$) |
| PLAD | 0.09 | 0.13 | 0.12 | 0.16 | 0.21 | 0.22 |
| Shape2Prog | 0.03 | 0.46 | 0.06 | 0.06 | 0.25 | 0.40 |
| Dataset Matches | 0.25 | 0.11 | 0.08 | 0.17 | 0.08 | 0.41 |
| MeshCoder | 0.005 | 0.52 | 0.011 | 0.61 | 0.0015 | 0.78 |

Note: "Dataset Matches" represent examples from the training set with minimal CD loss to Trellis-generated point clouds.

**Table 10: Reconstruction metrics under different point cloud quantities**

| Number of Points | CD Loss ($\downarrow$) | IoU ($\uparrow$) |
|---|---|---|
| 4096 | 0.000404 | 0.8707 |
| 8192 | 0.000396 | 0.8718 |
| 16384 | 0.000325 | 0.8907 |

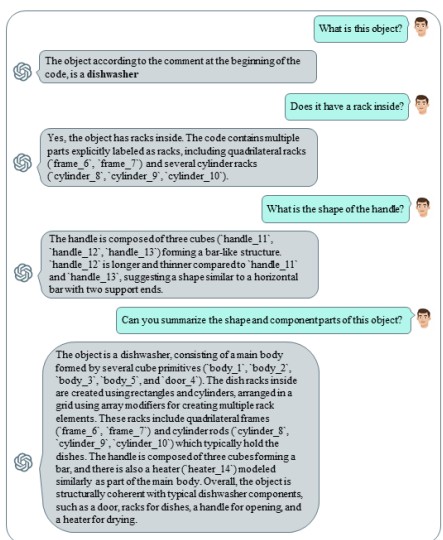

Figure 19: Experiments on how GPT-4o can understand shape through given dishwasher code.

```python
import bpy
from math import radians, pi
from bpy_lib import *

delete_all()

# object name: office chair
# part_1: wheel
create_primitive(name='wheel_1',
primitive_type='cylinder',
location=...)
bevel(name='wheel_1', ...)

# part_2: wheel
create_primitive(...)

# part_3: wheel
create_primitive(...)
bevel(...)

# part_4: wheel
create_primitive(...)

# part_5: wheel cap
create_rectangle(...)
create_arc_by_3Dpoints(...)

# part_6: wheel axle
create_primitive(...)

# part_7: wheel cap
create_rectangle(...)
create_arc_by_3Dpoints(...)

# part_8: wheel axle
create_primitive(...)

# part_9: wheel axle
create_primitive(...)

# part_10: wheel cap
create_rectangle(...)
create_arc_by_3Dpoints(...)

# part_11: wheel cap
create_rectangle(...)
create_arc_by_3Dpoints(...)

# part_12: wheel axle
create_primitive(...)

# part_13: chair base
create_curve(...)
bevel(...)
create_curve(...)

# part_14: chair base
create_curve(name='curve_14',
control_points=[[0.0, 0.0, 0.0],
[0.0, -0.025, 0.0], [0.044, -
0.025, 0.0], [0.044, 0.0, 0.0]],
handle_type=[1, 1, 1, 1, 1, 1,
1, 1], closed=True)
bevel(...)
create_curve(...)
```

```python
# part_15: chair base
create_curve(name='curve_15',
control_points=[[0.0, 0.0, 0.0],
[-0.0, 0.025, 0.0], [-0.044,
0.025, 0.0], [-0.044, -0.0,
0.0]], handle_type=[1, 1, 1, 1,
1, 1, 1, 1], closed=True)
bevel(...)
create_curve(...)

# part_16: chair base
create_curve(name='curve_16',
control_points=[[0.0, 0.0, 0.0],
[0.0, 0.025, 0.0], [-0.044,
0.025, 0.0], [-0.044, 0.0,
0.0]], handle_type=[1, 1, 1, 1,
1, 1, 1, 1], closed=True)
bevel(...)
create_curve(...)

# part_17: leg
create_primitive(...)

# part_18: leg
create_primitive(...)

# part_19: seat
create_circle(name=['Circle_0_1
9', 'Circle_1_19',
'Circle_2_19', 'Circle_3_19',
'Circle_4_19', 'Circle_5_19',
'Circle_6_19', 'Circle_7_19',
'Circle_8_19'], location=[...])
bridge_edge_loops(name=...)
join_obj(name='seat_19',
seq_name=...,
weld_threshold=...)
add_simple_deform_modifier(name
='seat_19', angle=...,
origin=..., rotation=...)
create_curve(name='curve_19',
control_points=...,
points_radius=...,
handle_type=...)
add_curve_modifier_to_object(na
me='seat_19',
curve_name='curve_19',
origin=..., rotation=...,
axis='POS_Z')
```

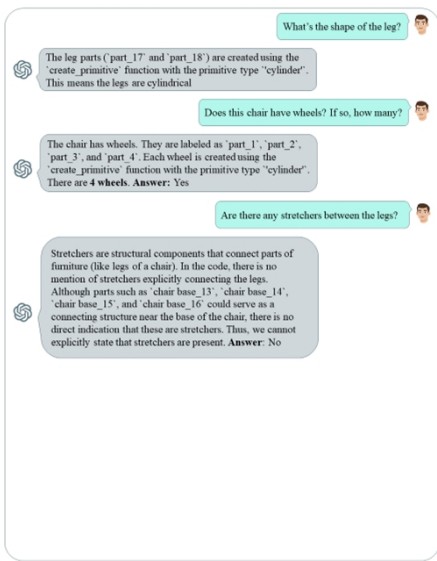

> What's the shape of the leg?

The leg parts (`part_17` and `part_18`) are created using the `create_primitive` function with the primitive type "cylinder". This means the legs are cylindrical

> Does this chair have wheels? If so, how many?

The chair has wheels. They are labeled as `part_1`, `part_2`, `part_3`, and `part_4`. Each wheel is created using the `create_primitive` function with the primitive type "cylinder". There are **4 wheels**. **Answer:** Yes

> Are there any stretchers between the legs?

Stretchers are structural components that connect parts of furniture (like legs of a chair). In the code, there is no mention of stretchers explicitly connecting the legs. Although parts such as `chair base_13`, `chair base_14`, `chair base_15`, and `chair base_16` could serve as a connecting structure near the base of the chair, there is no direct indication that these are stretchers. Thus, we cannot explicitly state that stretchers are present. **Answer**: No

Figure 20: Experiments on how GPT-4o can understand shape through given office chair code.

Figure 21: Experiments on how GPT-4o can understand shape through given cell shelf code.

```python
"""
Create primitive object
"""
def create_primitive(name, primitive_type="cube", location=None, scale=None, rotation=None, rotation_mode='QUATERNION', apply=False,
x_subdivisions=None, y_subdivisions=None, use_minimum_face=USE_MINIMUM_FACE, average_edge_length=AVERAGE_EDGE_LENGTH, resolution=None):

    if primitive_type=="uv_sphere":
        if not use_minimum_face:
            if average_edge_length !=None:
                res=int(2*pi//average_edge_length)
                segments, ring_count=res, res
            else:
                segments, ring_count=resolution[0], resolution[1]
            getattr(bpy.ops.mesh, f"primitive_{primitive_type}_add")(segments=segments,ring_count=ring_count)
            primitive = bpy.context.object
            primitive.name = name
        else:
            getattr(bpy.ops.mesh, f"primitive_{primitive_type}_add")()
            primitive = bpy.context.object
            primitive.name = name

    if primitive_type in ["cylinder","cone"]:
        if not use_minimum_face:
            if average_edge_length !=None:
                res_1=int(2*pi//average_edge_length)
                vertices=res_1
                res_2=int(2//average_edge_length)
            else:
                vertices,res_2=resolution[0], resolution[1]
            getattr(bpy.ops.mesh, f"primitive_{primitive_type}_add")(vertices=vertices)
            primitive = bpy.context.object
            primitive.name = name
            if primitive_type=="cylinder":
                primitive=subdivide_primitive(name,[res_2],['Z'])
            else:
                primitive=split_cone_z(name,res_2)
        else:
            getattr(bpy.ops.mesh, f"primitive_{primitive_type}_add")()
            primitive = bpy.context.object
            primitive.name = name

    if primitive_type=="torus":
        if not use_minimum_face:
            if average_edge_length !=None:
                res_1=int(2*pi//average_edge_length)
                res_2=int(0.5*pi//average_edge_length)
            else:
                res_1, res_2=resolution[0], resolution[1]
            getattr(bpy.ops.mesh, f"primitive_{primitive_type}_add")(major_segments=res_1,minor_segments=res_2)
            primitive = bpy.context.object
            primitive.name = name
        else:
            getattr(bpy.ops.mesh, f"primitive_{primitive_type}_add")()
            primitive = bpy.context.object
            primitive.name = name

    if primitive_type=="cube":
        if not use_minimum_face:
            if average_edge_length !=None:
                res=int(2//average_edge_length)
                if res>1:
                    resolution=[res,res,res]
            else:
                pass
            getattr(bpy.ops.mesh, f"primitive_{primitive_type}_add")()
            primitive = bpy.context.object
            primitive.name = name
            primitive=subdivide_primitive(name,resolution,['X','Y','Z'])
        else:
            getattr(bpy.ops.mesh, f"primitive_{primitive_type}_add")()
            primitive = bpy.context.object
            primitive.name = name

    if primitive_type=="grid":
        getattr(bpy.ops.mesh, f"primitive_{primitive_type}_add")(x_subdivisions=x_subdivisions,y_subdivisions=y_subdivisions)
        primitive = bpy.context.object
        primitive.name = name

    if location:
        primitive.location = location
    if scale:
        primitive.scale = scale
    if rotation:
        if rotation_mode=='XYZ':
            primitive.rotation_euler = [angle * pi for angle in rotation]
        elif rotation_mode=='QUATERNION':
            primitive.rotation_mode = 'QUATERNION'
            primitive.rotation_quaternion = rotation
        elif rotation_mode=='MATRIX':
            mat = np.eye(4)
            rotation = np.array(rotation).reshape([3,3])
            mat[:3,:3] = rotation
            bpy.context.view_layer.update()
            world_matrix = torch.tensor(bpy.data.objects[name].matrix_world)
            scale_now = world_matrix.norm(dim=0)[:3]
            scale_matrix = torch.eye(4)
            scale_matrix[0,0],scale_matrix[1,1],scale_matrix[2,2] =scale_now[0],scale_now[1],scale_now[2]
            scale_matrix_inv = scale_matrix.clone()
            for i in range(3):
                if scale_matrix_inv[i,i]>1e-10:
                    scale_matrix_inv[i,i]=1.0 / scale_matrix_inv[i,i]
            mat = scale_matrix_inv@torch.tensor(mat,dtype=torch.float32)@scale_matrix
            mat = mathutils.Matrix(np.array(mat))
            bpy.data.objects[name].matrix_world = bpy.data.objects[name].matrix_world@mat

    if apply:
        bpy.ops.object.transform_apply(location=True, rotation=True, scale=True)

    return primitive
```

Figure 22: Implementation of the function for creating primitives

```python
"""
Creates a translational object of a line trajectory
"""
def create_curve(name, profile_name=None,control_points=[],points_radius=[],handle_type=[],closed=False, center="POINT",thickness=None,
fill_caps="none",flip_normals=False, bevel_width=None, bevel_segments=8,use_minimum_face=USE_MINIMUM_FACE, average_edge_length=AVERAGE_EDGE_LENGTH, resolution=24,
volumn_origin=True):
    if isinstance(name, str):
        type_dict={0:"AUTO", 1:"VECTOR", 2:"ALIGNED", 3:"FREE"}

        control_points = np.array(control_points).tolist()
        control_points_tmp = copy.deepcopy(control_points)
        num_handle_co = handle_type.count(3)
        num_control_points = len(control_points) - num_handle_co

        curveData = bpy.data.curves.new(name, type='CURVE')
        curveData.dimensions = '3D'

        bezierSpline = curveData.splines.new('BEZIER')
        bezierSpline.bezier_points.add(num_control_points - 1)
        bezierSpline.use_cyclic_u = closed

        for i in range(num_control_points):
            bezier_point = bezierSpline.bezier_points[i]
            bezier_point.handle_left_type = type_dict[handle_type[2*i]]
            if type_dict[handle_type[2*i]]=="FREE":
                bezier_point.handle_left = control_points.pop(0)
            bezier_point.co = control_points.pop(0)
            bezier_point.handle_right_type = type_dict[handle_type[2*i+1]]
            if type_dict[handle_type[2*i+1]]=="FREE":
                bezier_point.handle_right = control_points.pop(0)
            bezier_point.radius = points_radius[i] if len(points_radius)!=0 else 1.0

        assert len(control_points)==0, "cannot create curve"
        if use_minimum_face:
            use_resolution = 12
        elif not average_edge_length is None:
            for i in range(len(bezierSpline.bezier_points) - 1):
                p1 = bezierSpline.bezier_points[i].co
                p2 = bezierSpline.bezier_points[i + 1].co
                total_length += (p2 - p1).length
            use_resolution = total_length/average_edge_length

        if resolution:
            use_resolution = resolution
        curveData.resolution_u = use_resolution

        curveOB = bpy.data.objects.new(name, curveData)

        if profile_name != None:
            curveData.bevel_mode = "OBJECT"
            curveData.splines[0].use_smooth = False
            scn = bpy.context.scene.collection
            scn.objects.link(curveOB)

            if bevel_width!=None:
                bevel(name=name, width=bevel_width, segments=bevel_segments)
                curveData = bpy.data.objects[name].data
                curveData.bevel_mode = 'OBJECT'

            curveData.bevel_object = bpy.data.objects[profile_name]
            if fill_caps=="both":
                curveData.use_fill_caps = True
            else:
                curveData.use_fill_caps = False

            if use_minimum_face:
                use_resolution = 24
            elif not average_edge_length is None:
                for i in range(len(bezierSpline.bezier_points) - 1):
                    p1 = bezierSpline.bezier_points[i].co
                    p2 = bezierSpline.bezier_points[i + 1].co
                    total_length += (p2 - p1).length
                use_resolution = total_length/average_edge_length

            if resolution:
                use_resolution = resolution

            curveData.resolution_u = use_resolution

            bpy.context.view_layer.objects.active = bpy.data.objects[name]
            bpy.ops.object.mode_set(mode = 'OBJECT')
            bpy.data.objects[name].select_set(True)
            bpy.ops.object.convert(target='MESH')
            bpy.data.objects.remove(bpy.data.objects[profile_name], do_unlink=True)
            if volumn_origin:
                bpy.ops.object.origin_set(type='ORIGIN_CENTER_OF_VOLUME', center='MEDIAN')

            if fill_caps in ["start","end"]:
                make_caps(name,fill_caps)

            if flip_normals:
                recalculate_normals(name,inside=True)
            else:
                recalculate_normals(name,inside=False)

            if thickness>1e-10:
                solidify(name,thickness)

            weld(name,1e-5)

            return curveOB
        else:
            scn = bpy.context.scene.collection
            scn.objects.link(curveOB)
            if center=="MEDIAN":
                bpy.data.objects[name].select_set(True)
                bpy.ops.object.origin_set(type='ORIGIN_GEOMETRY', center='MEDIAN')
            points=np.array(control_points_tmp)
            return {"name":name, "points":points, "handle_type":handle_type, "closed":closed, "center":center}

    elif profile_name==None:
        if isinstance(profile_name, str):
            profile_name = [profile_name]*len(name)
        if len(points_radius) != 0 and (isinstance(points_radius[0], float) or isinstance(points_radius[0], int)):
            points_radius = [points_radius]*len(name)
        elif len(points_radius) == 0:
            points_radius = [[]] * len(name)
        if isinstance(handle_type[0], int):
            handle_type = [handle_type]*len(name)
        if isinstance(closed, bool):
            closed = [closed]*len(name)
        if isinstance(center, str):
            center = [center]*len(name)
        for i in range(len(name)):
            create_curve(name=name[i], control_points=control_points[i], points_radius=points_radius[i], handle_type=handle_type[i], closed=closed[i],
center=center[i], thickness=thickness, fill_caps=fill_caps, flip_normals=flip_normals, resolution=resolution)
        points = np.array(copy.deepcopy(control_points))
        return {"name":name, "points":points, "handle_type":handle_type, "closed":closed, "center":center}
```

Figure 23: Implementation of the function for creating curves

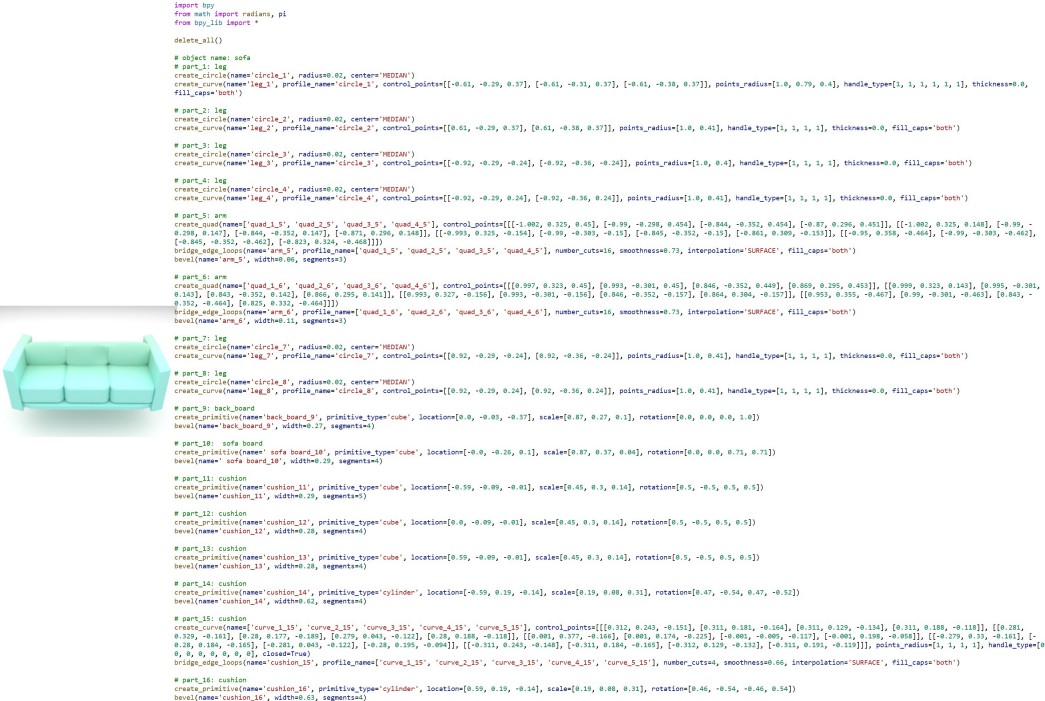

Figure 24: An example of sofa. The input is a point cloud of a sofa, and the figure shows the code inferred by the object-to-code inference model, as well as the resulting mesh generated by executing the inferred code.

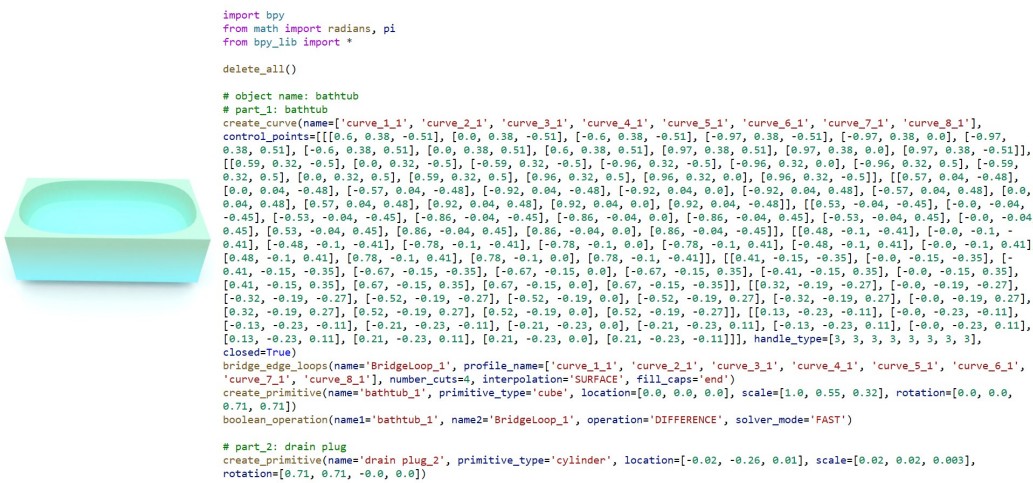

Figure 25: An example of bathtub. The input is a point cloud of a bathtub, and the figure shows the code inferred by the object-to-code inference model, as well as the resulting mesh generated by executing the inferred code.

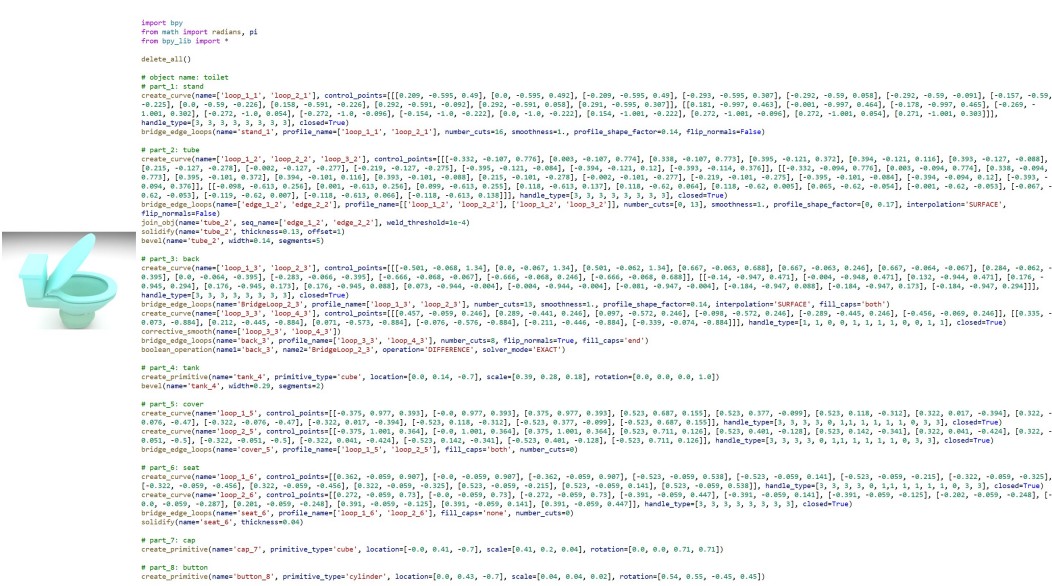

```
import bpy
from math import radians, pi
from bpy_lib import *

delete_all()

# object name: toilet
# part_1: stand
create_curve(name=['loop_1_1', 'loop_2_1'], control_points=[[[0.209, -0.595, 0.49], [0.0, -0.595, 0.492], [-0.209, -0.595, 0.49], [-0.293, -0.595, 0.307], [-0.292, -0.59, 0.058], [-0.292, -0.59, -0.091], [-0.157, -0.59,
-0.225], [0.0, -0.59, -0.226], [0.158, -0.591, -0.226], [0.292, -0.591, -0.092], [0.292, -0.591, 0.058], [0.291, -0.595, 0.307]], [[0.181, -0.997, 0.463], [-0.001, -0.997, 0.464], [-0.178, -0.997, 0.465], [-0.269, -
1.001, 0.302], [-0.272, -1.0, 0.054], [-0.272, -1.0, -0.096], [-0.154, -1.0, -0.222], [0.0, -1.0, -0.222], [0.154, -1.001, -0.222], [0.272, -1.001, -0.096], [0.272, -1.001, 0.054], [0.271, -1.001, 0.303]]],
handle_type=[3, 3, 3, 3, 3, 3, 3], closed=True)
bridge_edge_loops(name='stand_1', profile_name=['loop_1_1', 'loop_2_1'], number_cuts=16, smoothness=1., profile_shape_factor=0.14, flip_normals=False)

# part_2: tube
create_curve(name=['loop_1_2', 'loop_2_2', 'loop_3_2'], control_points=[[[-0.332, -0.107, 0.776], [0.003, -0.107, 0.774], [0.338, -0.107, 0.773], [0.395, -0.121, 0.372], [0.394, -0.121, 0.116], [0.393, -0.127, -0.088],
[0.215, -0.127, -0.278], [-0.002, -0.127, -0.277], [-0.219, -0.127, -0.275], [-0.395, -0.121, -0.084], [-0.394, -0.121, 0.12], [-0.393, -0.114, 0.376]], [[-0.332, -0.094, 0.776], [0.003, -0.094, 0.774], [0.338, -0.094,
0.773], [0.395, -0.101, 0.372], [0.394, -0.101, 0.116], [0.393, -0.101, -0.088], [0.215, -0.101, -0.278], [-0.002, -0.101, -0.277], [-0.219, -0.101, -0.275], [-0.395, -0.101, -0.084], [-0.394, -0.094, 0.12], [-0.393, -
0.094, 0.376]], [[-0.098, -0.613, 0.256], [0.001, -0.613, 0.256], [0.099, -0.613, 0.255], [0.118, -0.613, 0.137], [0.118, -0.62, 0.064], [0.118, -0.62, 0.005], [0.065, -0.62, -0.054], [-0.001, -0.62, -0.053], [-0.067, -
0.62, -0.053], [-0.119, -0.62, 0.007], [-0.118, -0.613, 0.066], [-0.118, -0.613, 0.138]]], handle_type=[3, 3, 3, 3, 3, 3, 3], closed=True)
bridge_edge_loops(name=['edge_1_2', 'edge_2_2'], profile_name=['loop_1_2', 'loop_2_2'], ['loop_1_2', 'loop_3_2']], number_cuts=[0, 13], smoothness=1., profile_shape_factor=[0, 0.17], interpolation='SURFACE',
flip_normals=False)
join_obj(name='tube_2', seq_name=['edge_1_2', 'edge_2_2'], weld_threshold=1e-4)
solidify(name='tube_2', thickness=0.13, offset=1)
bevel(name='tube_2', width=0.14, segments=5)

# part_3: back
create_curve(name=['loop_1_3', 'loop_2_3'], control_points=[[[-0.501, -0.068, 1.34], [0.0, -0.067, 1.34], [0.501, -0.062, 1.34], [0.667, -0.063, 0.688], [0.667, -0.063, 0.246], [0.667, -0.064, -0.067], [0.284, -0.062, -
0.395], [0.0, -0.064, -0.395], [-0.285, -0.066, -0.395], [-0.666, -0.068, -0.067], [-0.666, -0.068, 0.246], [-0.666, -0.068, 0.688]], [[-0.14, -0.947, 0.471], [-0.004, -0.948, 0.471], [0.132, -0.944, 0.471], [0.176, -
0.945, 0.294], [0.176, -0.945, 0.173], [0.176, -0.945, 0.088], [0.073, -0.944, -0.004], [-0.004, -0.944, -0.004], [-0.081, -0.947, -0.004], [-0.184, -0.947, 0.088], [-0.184, -0.947, 0.173], [-0.184, -0.947, 0.294]]],
handle_type=[3, 3, 3, 3, 3, 3, 3], closed=True)
bridge_edge_loops(name='BridgeLoop_2_3', profile_name=['loop_1_3', 'loop_2_3'], number_cuts=13, smoothness=1., profile_shape_factor=0.14, interpolation='SURFACE', fill_caps='both')
create_curve(name=['loop_3_3', 'loop_4_3'], control_points=[[[0.457, -0.059, 0.246], [0.289, -0.441, 0.246], [0.097, -0.572, 0.246], [-0.098, -0.572, 0.246], [-0.289, -0.445, 0.246], [-0.456, -0.069, 0.246]], [[0.335, -
0.073, -0.884], [0.212, -0.445, -0.884], [0.071, -0.573, -0.884], [-0.076, -0.576, -0.884], [-0.211, -0.446, -0.884], [-0.339, -0.074, -0.884]]], handle_type=[1, 1, 0, 0, 1, 1, 1, 1, 0, 0, 1, 1], closed=True)
corrective_smooth(name=['loop_3_3', 'loop_4_3'])
bridge_edge_loops(name='back_3', profile_name=['loop_3_3', 'loop_4_3'], number_cuts=8, flip_normals=True, fill_caps='end')
boolean_operation(name1='back_3', name2='BridgeLoop_2_3', operation='DIFFERENCE', solver_mode='EXACT')

# part_4: tank
create_primitive(name='tank_4', primitive_type='cube', location=[0.0, 0.14, -0.7], scale=[0.39, 0.28, 0.18], rotation=[0.0, 0.0, 0.0, 1.0])
bevel(name='tank_4', width=0.29, segments=2)

# part_5: cover
create_curve(name='loop_1_5', control_points=[[-0.375, 0.977, 0.393], [-0.0, 0.977, 0.393], [0.375, 0.977, 0.393], [0.523, 0.687, 0.155], [0.523, 0.377, -0.099], [0.523, 0.118, -0.312], [0.322, 0.017, -0.394], [0.322, -
0.076, -0.47], [-0.322, -0.076, -0.47], [-0.322, 0.017, -0.394], [-0.523, 0.118, -0.312], [-0.523, 0.377, -0.099], [-0.523, 0.687, 0.155]], handle_type=[3, 3, 3, 3, 0, 1,1, 1, 1, 1, 0, 3, 3], closed=True)
create_curve(name='loop_2_5', control_points=[[-0.375, 1.001, 0.364], [-0.0, 1.001, 0.364], [0.375, 1.001, 0.364], [0.523, 0.711, 0.126], [0.523, 0.401, -0.128], [0.523, 0.142, -0.341], [0.322, 0.041, -0.424], [0.322, -
0.051, -0.5], [-0.322, -0.051, -0.5], [-0.322, 0.041, -0.424], [-0.523, 0.142, -0.341], [-0.523, 0.401, -0.128], [-0.523, 0.711, 0.126]], handle_type=[3, 3, 3, 3, 0, 1,1, 1, 1, 1, 0, 3, 3], closed=True)
bridge_edge_loops(name='cover_5', profile_name=['loop_1_5', 'loop_2_5'], fill_caps='both', number_cuts=0)

# part_6: seat
create_curve(name='loop_1_6', control_points=[[0.362, -0.059, 0.007], [-0.0, -0.059, 0.007], [-0.362, -0.059, 0.007], [-0.523, -0.059, 0.538], [-0.523, -0.059, 0.141], [-0.523, -0.059, -0.215], [-0.322, -0.059, -0.325],
[-0.322, -0.059, -0.456], [0.322, -0.059, -0.456], [0.322, -0.059, -0.325], [0.523, -0.059, -0.215], [0.523, -0.059, 0.141], [0.523, -0.059, 0.538]], handle_type=[3, 3, 3, 3, 0, 1,1, 1, 1, 1, 1, 0, 3, 3], closed=True)
create_curve(name='loop_2_6', control_points=[[0.272, -0.059, 0.73], [-0.0, -0.059, 0.73], [-0.272, -0.059, 0.73], [-0.391, -0.059, 0.447], [-0.391, -0.059, 0.141], [-0.391, -0.059, -0.125], [-0.202, -0.059, -0.248], [-
0.0, -0.059, -0.287], [0.201, -0.059, -0.248], [0.391, -0.059, -0.125], [0.391, -0.059, 0.141], [0.391, -0.059, 0.447]], handle_type=[3, 3, 3, 3, 3, 3, 3], closed=True)
bridge_edge_loops(name='seat_6', profile_name=['loop_1_6', 'loop_2_6'], fill_caps='none', number_cuts=0)
solidify(name='seat_6', thickness=0.04)

# part_7: cap
create_primitive(name='cap_7', primitive_type='cube', location=[-0.0, 0.41, -0.7], scale=[0.41, 0.2, 0.04], rotation=[0.0, 0.0, 0.71, 0.71])

# part_8: button
create_primitive(name='button_8', primitive_type='cylinder', location=[0.0, 0.43, -0.7], scale=[0.04, 0.04, 0.02], rotation=[0.54, 0.55, -0.45, 0.45])
```

Figure 26: An example of toilet. The input is a point cloud of a toilet, and the figure shows the code inferred by the object-to-code inference model, as well as the resulting mesh generated by executing the inferred code.

```python
import bpy
from math import radians, pi
from bpy_lib import *

delete_all()

# object name: chair
# part_1: leg
create_primitive(name='leg_1', primitive_type='cube', location=[-0.44, -0.46, 0.37], scale=[0.53, 0.05, 0.05], rotation=[0.5, 0.51, -0.51, -0.49])
bevel(name='leg_1', width=0.12, segments=8)

# part_2: leg
create_primitive(name='leg_2', primitive_type='cube', location=[-0.31, -0.46, -0.46], scale=[0.53, 0.05, 0.05], rotation=[0.51, -0.5, -0.49, 0.51])

# part_3: leg
create_primitive(name='leg_3', primitive_type='cube', location=[0.31, -0.46, -0.46], scale=[0.53, 0.05, 0.05], rotation=[0.51, -0.5, -0.49, 0.51])

# part_4: leg
create_primitive(name='leg_4', primitive_type='cube', location=[0.44, -0.46, 0.37], scale=[0.53, 0.05, 0.05], rotation=[0.5, 0.51, -0.51, -0.49])

# part_5: leg decoration
create_primitive(name='leg decoration_5', primitive_type='cube', location=[-0.37, -0.35, -0.05], scale=[0.42, 0.05, 0.05], rotation=[0.76, 0.01, 0.65, -0.01])
bevel(name='leg decoration_5', width=0.13, segments=1)

# part_6: leg decoration
create_primitive(name='leg decoration_6', primitive_type='cube', location=[0.37, -0.35, -0.05], scale=[0.42, 0.05, 0.05], rotation=[0.65, -0.01, 0.76, 0.01])
bevel(name='leg decoration_6', width=0.15, segments=6)

# part_7: seat
create_curve(name='seat_7', control_points=[[0.0, 0.03, -0.51], [-0.35, 0.03, -0.51], [-0.47, 0.05, 0.21], [-0.49, 0.03, 0.41], [0.0, 0.03, 0.5], [0.49, 0.03, 0.41], [0.47, 0.05, 0.21], [0.35, 0.03, -0.51], [0.0, 0.03, -0.51]], handle_type=[0.0, 0.0, 1.0, 1.0, 0.0, 0.0, 0.0, 0.0, 0.0, 0.0, 0.0, 0.0, 0.0, 0.0, 1.0, 1.0, 0.0, 0.0])
fill_grid(name='seat_7', thickness=0.1042)
bevel(name='seat_7', width=0.05, segments=1)

# part_8: arm
create_circle(name='circle_8', radius=0.06, center='MEDIAN')
create_curve(name='arm_8', profile_name='circle_8', control_points=[[-0.33, 0.64, -0.46], [-0.39, 0.64, -0.2], [-0.46, 1.02, -0.06], [-0.46, 0.64, 0.21], [-0.46, 0.1, 0.31]], points_radius=[1.0, 1.0, 1.0], handle_type=[0, 3, 3, 1, 1, 0], thickness=0.001, fill_caps='both')

# part_9: back
create_primitive(name='back_9', primitive_type='cube', location=[-0.31, 0.54, -0.46], scale=[0.45, 0.04, 0.04], rotation=[0.5, 0.51, -0.51, -0.49])

# part_10: back
create_primitive(name='back_10', primitive_type='cube', location=[0.31, 0.54, -0.46], scale=[0.45, 0.04, 0.04], rotation=[0.5, 0.51, 0.51, 0.49])

# part_11: arm
create_circle(name='circle_11', radius=0.06, center='MEDIAN')
create_curve(name='arm_11', profile_name='circle_11', control_points=[[0.33, 0.64, -0.46], [0.39, 0.64, -0.21], [0.46, 1.01, -0.05], [0.47, 0.62, 0.23], [0.47, 0.09, 0.32]], points_radius=[1.0, 1.0, 1.0], handle_type=[0, 3, 3, 1, 1, 0], thickness=0.001, fill_caps='both')

# part_12: back decoration
create_curve(name='back decoration_12', control_points=[[-0.35, 0.8, -0.51], [-0.35, 0.62, -0.51], [0.01, 0.62, -0.51], [0.35, 0.62, -0.51], [0.35, 0.8, -0.51], [0.35, 0.99, -0.51], [0.01, 0.99, -0.51], [-0.35, 0.99, -0.51], [-0.35, 0.8, -0.51]], handle_type=[0.0, 0.0, 1.0, 1.0, 0.0, 0.0, 1.0, 1.0, 0.0, 0.0, 1.0, 1.0, 0.0, 0.0, 1.0, 1.0, 0.0, 0.0])
fill_grid(name='back decoration_12', thickness=0.086)
bevel(name='back decoration_12', width=0.04, segments=10)
```

```python
import bpy
from math import radians, pi
from bpy_lib import *

delete_all()

# object name: chair
# part_1: leg
create_primitive(name='leg_1', primitive_type='cube', location=[-0.44, -0.46, 0.37], scale=[0.53, 0.08, 0.08], rotation=[0.5, 0.51, -0.51, -0.49])
bevel(name='leg_1', width=0.12, segments=8)

# part_2: leg
create_primitive(name='leg_2', primitive_type='cube', location=[-0.31, -0.46, -0.46], scale=[0.53, 0.08, 0.08], rotation=[0.51, -0.5, -0.49, 0.51])

# part_3: leg
create_primitive(name='leg_3', primitive_type='cube', location=[0.31, -0.46, -0.46], scale=[0.53, 0.08, 0.08], rotation=[0.51, -0.5, -0.49, 0.51])

# part_4: leg
create_primitive(name='leg_4', primitive_type='cube', location=[0.44, -0.46, 0.37], scale=[0.53, 0.08, 0.08], rotation=[0.5, 0.51, -0.51, -0.49])

# part_5: leg decoration
create_primitive(name='leg decoration_5', primitive_type='cube', location=[-0.37, -0.35, -0.05], scale=[0.42, 0.05, 0.05], rotation=[0.76, 0.01, 0.65, -0.01])
bevel(name='leg decoration_5', width=0.13, segments=1)

# part_6: leg decoration
create_primitive(name='leg decoration_6', primitive_type='cube', location=[0.37, -0.35, -0.05], scale=[0.42, 0.05, 0.05], rotation=[0.65, -0.01, 0.76, 0.01])
bevel(name='leg decoration_6', width=0.15, segments=6)

# part_7: seat
create_curve(name='seat_7', control_points=[[0.0, 0.03, -0.51], [-0.35, 0.03, -0.51], [-0.47, 0.05, 0.21], [-0.49, 0.03, 0.41], [0.0, 0.03, 0.5], [0.49, 0.03, 0.41], [0.47, 0.05, 0.21], [0.35, 0.03, -0.51], [0.0, 0.03, -0.51]], handle_type=[0.0, 0.0, 1.0, 1.0, 0.0, 0.0, 0.0, 0.0, 0.0, 0.0, 0.0, 0.0, 0.0, 0.0, 1.0, 1.0, 0.0, 0.0])
fill_grid(name='seat_7', thickness=0.1042)
bevel(name='seat_7', width=0.05, segments=1)

# part_8: arm
create_circle(name='circle_8', radius=0.08, center='MEDIAN')
create_curve(name='arm_8', profile_name='circle_8', control_points=[[-0.33, 0.64, -0.46], [-0.39, 0.64, -0.2], [-0.46, 1.02, -0.06], [-0.46, 0.64, 0.21], [-0.46, 0.1, 0.31]], points_radius=[1.0, 1.0, 1.0], handle_type=[0, 3, 3, 1, 1, 0], thickness=0.001, fill_caps='both')

# part_9: back
create_primitive(name='back_9', primitive_type='cube', location=[-0.31, 0.54, -0.46], scale=[0.45, 0.04, 0.04], rotation=[0.5, 0.51, -0.51, -0.49])

# part_10: back
create_primitive(name='back_10', primitive_type='cube', location=[0.31, 0.54, -0.46], scale=[0.45, 0.04, 0.04], rotation=[0.5, 0.51, 0.51, 0.49])

# part_11: arm
create_circle(name='circle_11', radius=0.08, center='MEDIAN')
create_curve(name='arm_11', profile_name='circle_11', control_points=[[0.33, 0.64, -0.46], [0.39, 0.64, -0.21], [0.46, 1.01, -0.05], [0.47, 0.62, 0.23], [0.47, 0.09, 0.32]], points_radius=[1.0, 1.0, 1.0], handle_type=[0, 3, 3, 1, 1, 0], thickness=0.001, fill_caps='both')

# part_12: back decoration
create_curve(name='back decoration_12', control_points=[[-0.35, 0.8, -0.51], [-0.35, 0.62, -0.51], [0.01, 0.62, -0.51], [0.35, 0.62, -0.51], [0.35, 0.8, -0.51], [0.35, 0.99, -0.51], [0.01, 0.99, -0.51], [-0.35, 0.8, -0.51]], handle_type=[0.0, 0.0, 1.0, 1.0, 0.0, 0.0, 1.0, 1.0, 0.0, 0.0, 1.0, 1.0, 0.0, 0.0, 1.0, 1.0, 0.0, 0.0])
fill_grid(name='back decoration_12', thickness=0.086)
bevel(name='back decoration_12', width=0.04, segments=10)
```

Figure 27: By modifying the `scale` parameters of the `leg` and `arm` parts, we adjust their thickness. The highlighted sections indicate the changes made.

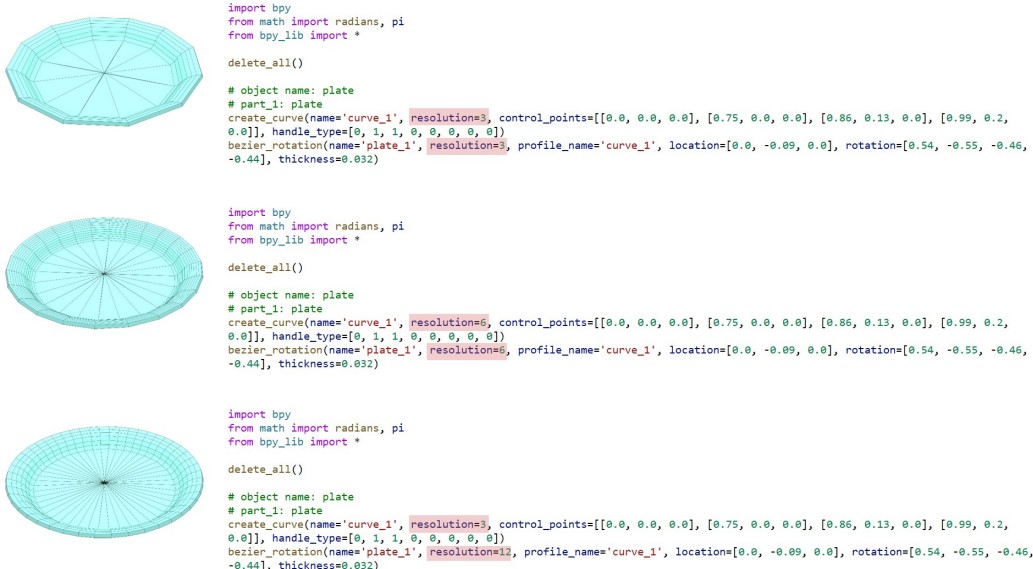

```
import bpy
from math import radians, pi
from bpy_lib import *

delete_all()

# object name: plate
# part_1: plate
create_curve(name='curve_1', resolution=3, control_points=[[0.0, 0.0, 0.0], [0.75, 0.0, 0.0], [0.86, 0.13, 0.0], [0.99, 0.2,
0.0]], handle_type=[0, 1, 1, 0, 0, 0, 0, 0])
bezier_rotation(name='plate_1', resolution=3, profile_name='curve_1', location=[0.0, -0.09, 0.0], rotation=[0.54, -0.55, -0.46,
-0.44], thickness=0.032)
```

```
import bpy
from math import radians, pi
from bpy_lib import *

delete_all()

# object name: plate
# part_1: plate
create_curve(name='curve_1', resolution=6, control_points=[[0.0, 0.0, 0.0], [0.75, 0.0, 0.0], [0.86, 0.13, 0.0], [0.99, 0.2,
0.0]], handle_type=[0, 1, 1, 0, 0, 0, 0, 0])
bezier_rotation(name='plate_1', resolution=6, profile_name='curve_1', location=[0.0, -0.09, 0.0], rotation=[0.54, -0.55, -0.46,
-0.44], thickness=0.032)
```

```
import bpy
from math import radians, pi
from bpy_lib import *

delete_all()

# object name: plate
# part_1: plate
create_curve(name='curve_1', resolution=3, control_points=[[0.0, 0.0, 0.0], [0.75, 0.0, 0.0], [0.86, 0.13, 0.0], [0.99, 0.2,
0.0]], handle_type=[0, 1, 1, 0, 0, 0, 0, 0])
bezier_rotation(name='plate_1', resolution=12, profile_name='curve_1', location=[0.0, -0.09, 0.0], rotation=[0.54, -0.55, -0.46,
-0.44], thickness=0.032)
```

Figure 28: By modifying the `resolution` parameter, we change its resolution. The highlighted sections indicate the changes made.

