# OpenReview forum: "MeshCoder: LLM-Powered Structured Mesh Code Generation from Point Clouds"
_NeurIPS.cc/2025/Conference — NeurIPS 2025 poster_

### Official Review · Reviewer_46UV · 2025-06-28

**Clarity:** 3
**Significance:** 2
**Originality:** 3
**Rating:** 4
**Confidence:** 4

**Summary:**

This paper introduces MeshLLM, a novel framework that leverages large language models (LLMs) to reconstruct 3D objects from point clouds into editable Blender Python scripts. The authors have designed a comprehensive set of APIs to model intricate geometries beyond simple primitives, enabling the synthesis of complex 3D structures. Then, a model trained to generate executable Blender scripts from point clouds, facilitating part-level reconstruction and intuitive editing through code modifications. MeshLLM outperforms existing shape-to-code methods in reconstruction tasks. The framework is evaluated properly, showcasing its ability to handle complex geometries and enable precise geometric and topological edits.

**Questions:**

(1) The paper acknowledges that MeshLLM is primarily designed for human-made objects and struggles with organic shapes (e.g., animals, humans). Given that many real-world applications (e.g., medical imaging, character modeling) require handling organic forms, how do the authors justify the framework’s broader applicability?
(2) In the mentioned pipeline, if there are certain errors in the code predicted by the mesh after the object is split into parts in Section 3.3, then this would be considered an erroneous sample when constructing the dataset. It seems that such errors cannot be corrected in subsequent training. How do the authors address this issue? Additionally, under the current settings, are there potential failure cases? Furthermore, do common LLM hallucination issues, such as repeatedly generating similar code, also exist here? The authors are kindly requested to elaborate on these points.

From the current submission format, it appears that the authors have placed greater emphasis on dataset construction, resulting in relatively limited theoretical innovation. Additionally, the ablation studies on various aspects of the model are somewhat insufficient. Therefore, I tentatively recommend rejection. However, if the authors can address the aforementioned issues, I am open to raising the score. Thanks.

**Ethical Concerns:**

["NO or VERY MINOR ethics concerns only"]

**Final Justification:**

Thanks for the rebuttal. The rebuttal did address my concerns, particularly regarding novelty and the ablation studies. I will raise my score to 'borderline accept'. Please make sure to include the necessary discussions in the revision.

**Limitations:**

Yes.

**Quality:**

3

**Strengths And Weaknesses:**

MeshLLM represents a notable advancement in 3D shape reconstruction and editing, offering a scalable and flexible solution for generating editable code from point clouds. Its strengths lie in its innovative use of LLMs, expressive APIs, and large-scale dataset construction. However, limitations such as restricted generalization to organic shapes and lack of open access to data/code may impact its broader applicability. A short summary is listed as follows:

** Strength
(1) It bridges the gap between 3D point clouds and editable code representations, offering a flexible solution for reverse engineering and shape editing.
(2) The Blender Python APIs support advanced geometric operations (e.g., Boolean operations, sweeping trajectories), enabling the modeling of intricate real-world objects.
(3) Quantitative results demonstrate significant improvements over baseline methods in reconstruction accuracy (e.g., higher IoU, lower Chamfer Distance).
(4) The code-based representation allows for intuitive geometric and topological modifications, enhancing usability in downstream applications.

** Weakness
(1) The framework primarily targets human-made objects and may not generalize well to organic shapes (e.g., animals, humans), as noted in the limitations. Besides, it is tightly coupled with Blender, which may restrict its applicability to other 3D modeling software or environments.
(2) While the paper mentions ablation studies, more detailed analysis of the impact of individual components (e.g., tokenizer design, LLM architecture) would strengthen the statement.

---

> ### Author Rebuttal · Authors · 2025-07-27
>
> **Thank you sincerely for your detailed review and valuable insights!** We greatly appreciate your recognition of MeshLLM’s strengths, including its innovation in bridging 3D point clouds and editable code, expressive Blender APIs, improved reconstruction accuracy, and enhanced usability through code-based editing. Your thoughtful feedback on limitations and questions is crucial for us to strengthen the completeness of our work. We address your concerns systematically below.
>
> **Response to the lack of open access to data/code**
> We fully acknowledge the importance of open access for reproducibility and community advancement. We are pleased to share that our function collection has already been open-sourced and can be found in the supplementary materials, specifically in the file `bpy_lib.py`. Additionally, upon acceptance of our paper, we commit to releasing the entire dataset, all source codes, and the model's training checkpoints to ensure full reproducibility and community access.
>
> **Response to the ability to generalize to organisms**
> While MeshLLM primarily targets human made objects, we would like to kindly clarify that we have not acknowledged that MeshLLM struggles with organic objects when discussing its limitations. We stated that the applicability of MeshLLM to organic objects is underdeveloped and we leave this as a future work. In fact, we believe MeshLLM has the potential to generalize to organic objects. To achieve this, we need to 1) collect a dataset of organic objects decomposed into constituent parts and 2) represent every part as code using our Blender APIs.
> 1)  Recently part-aware 3D generation works, such as PartCrafter[1] and FromOnetoMore[2], demonstrate that we can filter and process objects in Objaverse and obtain a wide range of objects decomposed into parts, which contains organic objects. In addition, Infinigen contains many procedural programs that can generate organic objects in semantically meaningful parts and can be used as training data. It is worth noting that Infinigen already demonstrates that it is feasible to generate organic objects such as animals using Blender codes.
> 2) We posit that most parts can be represented by our Blender APIs. For some simple parts, we can use Primitive and Translation tools to represent them. For complex compound parts that can't be constructed by Primitive and Translation tools, we can decompose them into simpler sub-parts which could be constructed using Primitive and Translation tools, and join them using Boolean operations to form complex parts. For complex monolithic parts that are difficult to decompose into simpler parts, we can almost always use Bridge Loop tool to contruct them. Specifically, we can cut the part along an appropriate polyline or Bezier curve at multiple points and obtain multiple cross-sectional contours. We can fit every contour using a Bezier curve. Finally, the part can be obtained by connecting these ordered section Bezier curves using the Bridge Loop tool.
> In addition, for a part with fine-grained details (e.g., animal feathers or scales) , we can first represent its coarse shape as code using our Blender APIs, and then train a neural network to predict offset of every vertex in the coarse mesh. In this way, we can restore details on the surface by deforming the coarse mesh. This is consistent with workflows used by most artists: They often start by creating a coarse base mesh from geometric primitives and then moving to sculpting tools (like ZBrush) to deform the base mesh and recover intricate features.
>
> **Response to the coupling with Blender**
> We appreciate the reviewer's point about the coupling with Blender. We chose Blender initially for its open-source nature, robust Python API, and suitability for programmatic 3D asset generation. Our core contribution—the code-based constructive methodology—is not Blender-dependent; it adapts readily to other scriptable 3D tools (e.g., CAD software) with shared core functionalities like sweep operations. While our current constructive code is Blender-specific, these functions can be re-implemented in other code-capable 3D software. By retaining consistent functions, the inferred code works directly in these environments. Thus, only our current functions are Blender-coupled—our trained model and constructive logic are adaptable to other tools. Notably, our framework supports interoperability by exporting to standard formats, enabling seamless use across 3D pipelines.
>
> **Response to ablation studies**
> Thank you for suggesting a more detailed ablation studies. We have already conducted several ablation studies. We would like to respectfully point reviewer to Appendix A.4.1, where we provide three detailed ablation studies on key design choices. These experiments already investigate:
>
> 1. The impact of the shape tokenizer's triplane resolution and the number of learnable tokens (Table 5).
> 2. The critical importance of our two-stage training strategy (initializing the object model from the part model checkpoint), which significantly improves performance (Table 6).
> 3. A key aspect of the tokenizer architecture, confirming that our use of learnable tokens as queries to aggregate global information is superior to a simpler direct projection of local features (Table 7).
>
> As suggested, we will enhance this section by adding a new, targeted ablation study on the tokenizer. Specifically, we compare our triplane-based tokenizer against an alternative approach that uses point cross-attention based tokenizer, which is common in recent works like Michelangelo[3] and Craftsman[4]. The results demonstrate that for our task the triplane-based tokenizer performs significantly better. This finding confirms that the proposed shape tokenizer is crucial for capturing the constructive and symbolic nature of 3D shapes, leading to superior reconstruction accuracy and editability.
>
> |Method|CD Loss ($\downarrow$)|IoU($\uparrow$)|
> |---|---|---|
> |triplane-based| **0.00069**| **0.8387**|
> |point cross attention based|0.00089|0.8367|
>
> **Response to errors in the code predicted and the hallucination issues**
> Yes, the model’s inference process does exhibit hallucinations, leading to execution errors in the predicted code. We have statistically analyzed this phenomenon: the proportion of erroneous code in part-level inference results is about 0.1%, while that in object-level inference results is 0.66%. The primary manifestations of hallucinations include unreasonable numerical parameter values and syntactic errors in generated code, with no occurrence of code repetition.
>
> **Response to bad case predicted in subsequent training**
> We consider two types of bad cases for part predictions: erroneous code (as previously discussed) and suboptimal reconstructions. As noted in Appendix A.1.3, we applied a strict Chamfer Distance (CD) threshold of **0.005** for each inferred part—parts failing to meet this threshold are classified as bad cases. Notably, even predictions exceeding this threshold often yield visually similar shapes, reflecting the model’s robust approximation capability. **When constructing the object-code pairs dataset, we only include objects for which all constituent parts are good cases.** Therefore, these bad cases do not impact subsequent training processes.
> Our evaluation rigorously uses the original Infinigen test set, which explicitly includes objects with bad-case parts to ensure comprehensive assessment. To further validate generalization, we partitioned the test set into two subsets: 1. "All Parts as Good Cases" (objects where all parts meet the CD threshold) and 2. "Objects with Bad-Case Parts" (objects containing at least one part exceeding the threshold). Performance metrics across both subsets are shown below. We observe that MeshLLM maintains reasonably good performance on "Objects with Bad-Case Parts," with no significant degradation in metrics—this is because even when a shape can't be reconstructed with precise accuracy, the model is still able to generate a visually similar shape.
>
> | Method| All Parts as Good Cases| |Objects with Bad-Case Parts| |
> |---|---|---|---|---|
> |  |CD Loss ($\downarrow$)|IoU ($\uparrow$)|CD Loss ($\downarrow$)|IoU ($\uparrow$)|
> | PLAD| 0.026| 0.481|0.031| 0.473|
> | Shape2Prog| 0.056 |0.443|0.062|0.467|
> | MeshLLM| **0.00054**| **0.867** | **0.00089** | **0.76**  |
>
> **Response to the theoretical innovation**
> We thank the reviewer for the feedback and the opportunity to clarify our contributions' scope and novelty. Our innovation lies not in a single methodological improvement, but in introducing a complete, novel paradigm for 3D content generation/reconstruction, built on three foundations:
> 1. A highly expressive programmatic representation for 3D shapes—structured, interpretable, and compositional—offering a new theoretical basis for geometric reasoning.
> 2. A novel two-stage annotation methodology to address the field's critical data bottleneck, using an expert part-to-code model to automatically label large-scale datasets, enabling scalable training of models like MeshLLM.
> 3. MeshLLM itself, which validates this framework by demonstrating that an LLM can learn this programmatic language to generate high-quality, editable 3D assets. This holistic framework—encompassing representation, data pipeline, and generative model—pioneers a new class of 3D generative models and opens avenues for future theoretical research.
>
> [1]Lin Y, Lin C, Pan P, et al. PartCrafter: Structured 3D Mesh Generation via Compositional Latent Diffusion Transformers[J]. arXiv preprint arXiv:2506.05573, 2025.
>
> [2]Dong S, Ding L, Chen X, et al. From One to More: Contextual Part Latents for 3D Generation[J]. arXiv preprint arXiv:2507.08772, 2025.
>
> [3]Zhao Z, Liu W, Chen X, et al. Michelangelo: Conditional 3d shape generation based on shape-image-text aligned latent representation[J]. Advances in neural information processing systems, 2023, 36: 73969-73982.

---

### Official Review · Reviewer_w67m · 2025-06-29

**Clarity:** 4
**Significance:** 3
**Originality:** 3
**Rating:** 5
**Confidence:** 4

**Summary:**

The paper introduces a new dataset consisting of 1 million objects spanning 40 classes. Each object is composed of multiple parts with codes represented as a parameterized function implemented using the Blender API. The whole object is represented as a concatenation of the individual part codes.

The authors use a two stage regime to train a model that predicts object codes from an input point cloud. In the first stage a LLM model is trained to predict part codes from the corresponding point clouds. For each object in the dataset its part codes are predicted through this model and concatenated to form the object codes. The model is then initialized from the first stage weights and fine-tuned on the full object codes and point clouds.

**Questions:**

1. I wonder how the model would perform if using point clouds from generative models as input such as Point-E [1]? This would demonstrate the model's generalization capabilities and extend it to generative applications using image or text.

2. Additional details regarding the generated object's mesh quality can be useful. Comparing additional metrics such as face count against human made meshes or other recent 3D generation methods can help determine the model's usefulness in applications like simulations.

3. For articulated objects in the dataset, I’m curious how the parts are defined and distributed and whether parts are semantically meaningful for articulation applications.

4. I am curious about the scalability of such approaches to more object categories in the future. For example, how many more functions would need to be required in order to represent a dataset like Objaverse [2]? And whether it is actually feasible for such an approach given the effort needed to design these functions.

I am willing to increase my score if these points are addressed in the rebuttal.

[1] Nichol, Alex, et al. "Point-e: A system for generating 3d point clouds from complex prompts." arXiv preprint arXiv:2212.08751 (2022).

[2] Deitke, Matt, et al. "Objaverse: A universe of annotated 3d objects." Proceedings of the IEEE/CVF conference on computer vision and pattern recognition. 2023.

**Ethical Concerns:**

["NO or VERY MINOR ethics concerns only"]

**Final Justification:**

The authors have addressed my concerns during the rebuttal. I have read other reviewers' questions and author responses, and found the authors adequately addressed those concerns as well. Most reviewers are also positive of the paper and I think it will have a strong impact in the program based 3D generation direction. As such I will raise my rating of the paper to 5 accept.

**Limitations:**

yes

**Quality:**

4

**Strengths And Weaknesses:**

## Strengths

The paper writing is clear and well organized, covering most details regarding code representation and model training. The dataset provided is also reasonably large with a somewhat limited number of classes. However, it is significant considering the smaller number of available datasets available for program based object generation. The reconstruction results are convincing, with the LLM producing better shapes compared to prior program based methods. The method also has potential for applications in simulations where the part representation can yield cleaner meshes compared to other 3D representations like SDF.

## Weaknesses

While the paper has demonstrated impressive reconstruction results. It is lacking concrete examples that can better show how it can be applied in real world applications.  However, I believe this can be addressed with minor revision. I listed more specific questions regarding this below.

---

> ### Author Rebuttal · Authors · 2025-07-27
>
> **Thank you sincerely for your thorough review and valuable feedback!** We greatly appreciate your recognition of our work’s strengths, including the clear organization of the paper, the significance of the dataset for program-based object generation, and the convincing reconstruction results. Your insightful questions provide important guidance for enhancing the completeness of our research, and we address them systematically below.
>
>
> **Response to Q1**
> Using point clouds generated by generative models as input to MeshLLM for mesh reconstruction is indeed one of the key application directions of our method.
>
> We are truly grateful for the reviewer’s incredibly insightful suggestion to explore Point-E for point cloud generation—your attention to refining our experimental design has greatly enriched the rigor of our work. Following this valuable input, we conducted extensive experiments with Point-E. Unfortunately, we found that the point clouds generated by Point-E are of relatively poor quality. Therefore, we turned to Trellis[5], a widely acclaimed and robust method for single-image mesh generation. Specifically, we collected 10 samples for each of the three categories (chairs, shelves, and bottles) from online platforms, processed all 30 collected images using Trellis, sampled point clouds from the high-fidelity meshes it generated, and then fed these point clouds into MeshLLM for reconstruction.
>
> To further demonstrate that our model does not merely memorize training data, we identified examples from our training dataset with the smallest CD loss relative to the generated point clouds. All metric results are presented in the table below:
>
> | Method         | Chairs         |              | Shelves        |              | Bottles        |              |
> |----------------|----------------|--------------|----------------|--------------|----------------|--------------|
> |                | CD Loss (↓)    | IoU (↑)      | CD Loss (↓)    | IoU (↑)      | CD Loss (↓)    | IoU (↑)      |
> | PLAD           | 0.09          | 0.13         | 0.12          | 0.16         | 0.21          | 0.22         |
> | Shape2Prog     | 0.03          | 0.46         | 0.06          | 0.06         | 0.25          | 0.40         |
> | Dataset Matches | 0.25          | 0.11         | 0.08          | 0.17         | 0.08          | 0.41         |
> | **MeshLLM**    | **0.005**      | **0.52**     | **0.011**      | **0.61**     | **0.0015**      | **0.78**     |
>
>
> We can observe that MeshLLM still achieves superior reconstruction results even when using point clouds generated by generative models. These results demonstrate that our model can serve as a critical component in converting real-world images into 3D digital assets, bridging the gap between visual inputs and structured 3D representations for practical applications. However, not all examples can be reconstructed well. We found that the reconstruction results are suboptimal when attempting to reconstruct objects from categories not present in the training dataset. To further strengthen generalization in future work, we plan to (1) expand our training data to cover a more diverse set of object categories and topologies, such as  Objaverse [4], and (2) explore advanced alignment techniques, such as Direct Preference Optimization (DPO)[1], which is adopted by Deepmesh [2], to fine-tune the LLM's ability to handle noisier and more varied inputs. We will add a discussion of these preliminary findings and future plans to the paper.
>
>
> **Response to Q2**
> Thank you for raising this important point. The suitability of our meshes for applications like simulation is one of our method's key advantages.
>
> A core feature of our programmatic representation is that mesh resolution is an explicit and editable parameter in the generated code. As we demonstrate in our shape editing examples (Section 4.4), users can easily modify the resolution parameter to generate meshes ranging from low-polygon to high-fidelity, depending on the application's needs. We show this explicitly in Figure 8 and Figure 28, where the density of the mesh is adjusted with a simple code change.
>
> As for mesh quality, unlike methods that rely on Marching Cubes over SDFs to produce dense, irregular triangular meshes, our approach generates structured geometry, often with clean quad-based topology. This structured quad topology is particularly well-suited for UV unwrapping, artistic editing, and so on as it simplifies these processes and reduces artifacts.
>
> To demonstrate our model's ability to create efficient digital assets, we compared the face counts of meshes generated by different methods across specific object categories. Specifically, we analyzed human-authored meshes from the ShapeNet dataset, meshes generated by one of the widely used 3D generation methods (e.g., Trellis[5]), and those produced by MeshLLM in default resolution. We selected four representative categories—chairs, sofas, bathtubs, and shelves—with 10 examples included for each category, and the quantitative results are presented in the table below. As shown in the table, the face counts of meshes generated by our method are close to those of human-authored assets, significantly lower than those generated by Trellis, and even lower than human-authored meshes in some categories.
>
> | Method               | Chair Face Count | Sofa Face Count | Bathtub Face Count | Shelf Face Count |
> |----------------------|------------------|-----------------|--------------------|------------------|
> | ShapeNet             | **1038**         | **1537**        | 14,527             | 239        |
> | Trellis              | 12,186           | 10,040          | 15,433             | 47,301            |
> | MeshLLM              | 5800             | 3115            | **3,625**          | **67**            |
>
>
> **Response to Q3**
> This is a crucial aspect and feasible future work of our framework. The part definitions in our dataset are semantically meaningful and directly suitable for articulation. Our methodology leverages the Infinigen Indoor dataset, where objects are procedurally generated with a semantically correct, hierarchical part structure. Our pipeline is designed to preserve this meaningful decomposition mostly. For instance, in our appendix, we show that an office chair is correctly inferred with distinct parts like 'wheel', 'wheel_axle', and 'chair_base' (Figure 20). This part-based semantic information is explicitly encoded in the comments of our generated code, making it immediately interpretable and usable for downstream applications that require object articulation. Actually, Infinite Mobility [3] follows the same segmentation as Infinigen, and are capable of producing articulated objects with semantically meaningful part label.
>
> **Response to Q4**
> We believe MeshLLM has the potential to generalize to more diverse datasets such as Objaverse. To represent objects in Objaverse as codes, we need to 1) decompose objects into constituent parts and 2) represent every part as code using our Blender APIs.
> 1)  Recently part-aware 3D generation works, such as PartCrafter [6] and From One to More [7], demonstrate that we can filter and process objects in Objaverse and obtain a wide range of objects decomposed into constituent parts, providing the necessary training data.
> 2) We posit that most parts can be represented by our Blender APIs. For some simple parts, we can use Primitive and Translation tools to represent them. For complex compound parts that can not be constructed by Primitive and Translation tools, we can decompose them into simpler sub-parts which could be constructed using Primitive and Translation tools, and join them using Boolean operations to form the complex compound part. For complex monolithic parts that are difficult to decompose into simpler parts, we can use Bridge Loop tool to contruct them. Specifically, we can cut the part along an appropriate polyline or Bezier curve at multiple points and obtain multiple cross-sectional contours. We can fit every contour using a planar Bezier curve. Finally, the part can be obtained by connecting these ordered section Bezier curves using the Bridge Loop tool.
>
> In addition, for a part with fine-grained details (e.g., animal feathers or scales) , we can first represent its coarse shape as code using our Blender APIs, and then train a neural network to predict offset of every vertex in the coarse mesh. In this way, we can restore details on the surface by deforming the coarse mesh. This is consistent with workflows used by most artists: They often start by creating a coarse base mesh from geometric primitives and then moving to sculpting tools (like ZBrush) to deform the base mesh and recover intricate surface features.
>
> [1] Rafailov, Rafael, et al. "Direct preference optimization: Your language model is secretly a reward model." Advances in neural information processing systems 36 (2023): 53728-53741.
> [2] Zhao, Ruowen, et al. "Deepmesh: Auto-regressive artist-mesh creation with reinforcement learning." arXiv preprint arXiv:2503.15265 (2025).
> [3] Lian X, Yu Z, Liang R, et al. Infinite Mobility: Scalable High-Fidelity Synthesis of Articulated Objects via Procedural Generation. arXiv preprint arXiv:2503.13424, 2025.
> [4] Deitke, Matt, et al. "Objaverse: A universe of annotated 3d objects." Proceedings of the IEEE/CVF conference on computer vision and pattern recognition. 2023.
> [5] Xiang, J., Lv, Z., Xu, S., Deng, Y., Wang, R., Zhang, B., Chen, D., Tong, X., & Yang, J. (2024). Structured 3D Latents for Scalable and Versatile 3D Generation. arXiv preprint arXiv:2412.01506.
> [6] Lin Y, Lin C, Pan P, et al. PartCrafter: Structured 3D Mesh Generation via Compositional Latent Diffusion Transformers. arXiv preprint arXiv:2506.05573, 2025.
> [7] Dong S, Ding L, Chen X, et al. From One to More: Contextual Part Latents for 3D Generation. arXiv preprint arXiv:2507.08772, 2025.

---

> > ### Comment · Reviewer_w67m · 2025-08-02
> >
> > Thanks to the authors for clarifying my concerns. I hope the additional results can be included in the revised version of the paper along with figures of the generation results from TRELLIS. I have read other reviewers' questions and author responses, and found the authors adequately addressed those concerns as well. I believe that the the paper will have strong impact on developing program based 3D object generation methods. As such, I will be raising my rating for the paper.

---

> > > ### Author Response · Authors · 2025-08-03
> > >
> > > Thank you sincerely for your positive feedback and the raised rating—your recognition of our work’s potential impact on program-based 3D object generation means a great deal to us.
> > >
> > > We fully acknowledge your suggestion and will include the figures of generation results from TRELLIS in the revised version of the paper to provide more intuitive comparisons. Additionally, we will incorporate the results of other supplementary experiments conducted during this rebuttal process, ensuring all new findings are clearly presented to strengthen the work.
> > >
> > > Once again, thank you for your invaluable guidance and support, which have been instrumental in enhancing the quality of our work.

---

### Official Review · Reviewer_4dat · 2025-07-02

**Clarity:** 2
**Significance:** 3
**Originality:** 3
**Rating:** 4
**Confidence:** 3

**Summary:**

MeshLLM aims to train an object-to-code inference model. To accomplish this, MeshLLM first trains a part-to-code inference model that predicts code for a single part. Then, the difficulty is increased by training a model that predicts code for every part of an object. Finally, the codes for every part of the object are concatenated together to represent all parts of the object. To enable model training, MeshLLM constructs a synthesized dataset of paired parts and their corresponding codes. MeshLLM achieves shape-to-code reconstruction tasks by developing a comprehensive set of expressive Blender Python APIs capable of synthesizing intricate geometries.

**Questions:**

For different objects, the number of points sampled by MeshLLM is fixed. For complex objects, such as those with many parts, will the tokenizer model's capability decrease? Thanks to the use of the triplane projection method, the model can project different numbers of sampled point clouds into a fixed length. Sampling different numbers of points may improve the model's capability, which could be a direction for future work.

**Ethical Concerns:**

["NO or VERY MINOR ethics concerns only"]

**Final Justification:**

Thanks to the authors' replies! My concerns have been addressed, and I will keep the positive score.

**Limitations:**

Yes.

**Quality:**

3

**Strengths And Weaknesses:**

**Strengths:**
1. The paper defines a series of shapes and corresponding algorithmic rules, enabling the expression of complex shapes in a highly efficient manner. For example, the "array" representation allows for the concise construction of structures where certain geometric shapes are repeated in specific patterns, using array methods to build the entire structure at once.
2. MeshLLM uses a triplane model to represent sparse point cloud structures as compact feature vectors, enabling an efficient tokenizer.
3. MeshLLM achieves improved reconstruction performance compared to previous methods in terms of Chamfer Distance loss and IoU. In addition to quantitative evaluation, qualitative evaluation also shows improved geometric quality.

**Weaknesses:**
1. Regarding shape understanding, since GPT is a closed-source model, it cannot be fine-tuned to understand the series of shapes and algorithmic rules defined during creation. Does directly inputting these codes and their function names help with 3D understanding, without providing the specific algorithmic rules, as shown in Fig. 19? Furthermore, MeshLLM claims to enhance LLMs’ understanding of 3D objects. It would be better to provide comparisons, such as giving GPT multi-view images of objects or other methods, and comparing their 3D understanding capabilities with those when given the codes in MeshLLM.
2. Minor weaknesses in visualization: Since MeshLLM is related to 3D objects, showing multi-view results of 3D objects or supplementary videos could better demonstrate the quality of the generated results.

---

> ### Author Rebuttal · Authors · 2025-07-27
>
> **Thank you sincerely for your thorough review and valuable feedback!** We greatly appreciate your recognition of MeshLLM’s strengths, including efficient shape expression, effective triplane-based tokenization, improved reconstruction performance, and your positive assessments of quality, significance, and originality. Your insights on weaknesses and questions are particularly helpful for enhancing our work’s clarity and rigor, and we address them systematically below:
>
>
> **Response to Weaknesses 1**
> In our shape understanding tasks, we first input the implementation details and functional descriptions of functions—specifically those used in the object's reconstruction code—to GPT, followed by the object's reconstruction code and specific questions. We apologize for not explaining clearly that Figure 19 currently only shows the reconstruction code and question segments; we will provide a more complete and clear demonstration of the full input process (including function implementations and descriptions) in the revised manuscript to better illustrate how we support GPT’s understanding of shape rules.
>
> In our further experiment, we ask GPT to answer questions given code, multiview (6 views from front, back, left, right, top, and bottom), and we found that the advantage of using code as input lies in: 1) Understanding the inner structure is very hard when providing images only. For instance, given multiview images GPT can see how many racks inside but it's hard to see how many bars between one rack, whereas given code GPT can easily obtain the number. 2) Providing a measure of the dimensions. For example, we asked GPT to answer the width of the table given absolute height width. In this case GPT was able to answer the correct width given code while it failed to measure the width if only the images provided. But giving multiview images has its own strength, such as identifying color and texture of the object while the code doesn't have any informations about the material. Additionally, we found that those questions that could not be answered by only providing multiview images could be correctly answered by providing both multiview images and code, and the accuracy in providing both images and code is higher than only providing either images or code, which showed that multiview images and code are complementary to each other in shape understanding tasks. We will provide more examples and results in the revised version to further demonstrate the code representation could enhance LLM's understanding of 3D assets.
>
>
> **Response to Weaknesses 2**
> We kindly thank the reviewer for raising this point, which is very crucial for solidifying the performance of MeshLLM. We will release more rendered images and videos surrounding the 3D assets generated by MeshLLM. We are sorry for not providing more visualization results here, as uploading images or videos is not permitted during the rebuttal phase. We will include these materials in the revised manuscript to provide a more comprehensive demonstration of geometric quality.
>
>
> **Response to Question**
> Thank you for your insightful question and suggestions. We need to clarify that, as stated in line 798 of our appendix, during the training phase, we do not use a fixed number of points. Instead, we randomly sample 4096 to 16385 points and add Gaussian noise for training. This approach, which we will emphasize more prominently in the revised manuscript’s main text, enhances the model's robustness and its ability to handle point clouds with different numbers of points, which aligns with the suggestions you put forward. Through our observations, when we use 16384 points for inference, a configuration consistent with the point count used for all reported metrics in our paper, we can reconstruct complex objects comprising nearly twenty components (see Figure 20 in the Appendix). Additionally, to investigate how varying point cloud densities affect reconstruction quality, we conducted the following supplementary experiments.
>
> Specifically, we designed experiments with three different point cloud quantities, where the number of input points for our trained MeshLLM was set to 4096, 8192, and 16384 respectively to observe its reconstruction metrics across four representative categories: bottles, cell shelves, chairs, and sofas. The metric results are shown in the table below. **Notably, we observe a clear trend: as the number of input points increases, the reconstruction quality consistently improves.**
>
> | Number of Points | CD Loss ($\downarrow$) | IoU ($\uparrow$) |
> |------------------|------------------------|------------------|
> | 4096             | 0.000404               | 0.8707           |
> | 8192             | 0.000396               | 0.8718           |
> | 16384            | $\mathbf{0.000325}$     | $\mathbf{0.8907}$ |
>
> *Table Y: Reconstruction metrics under different number of points.*
>
> We are grateful for your constructive feedback and will incorporate these improvements and clarifications into the revised manuscript.

---

> > ### Comment · Reviewer_4dat · 2025-08-05
> >
> > Thanks to the authors' replies! My concerns have been addressed, and I will keep the positive score.

---

> > > ### Author Response · Authors · 2025-08-06
> > >
> > > Thank you very much for your positive feedback and for taking the time to review our work. We appreciate your support and are glad that we were able to address your concerns.

---

### Official Review · Reviewer_WTgQ · 2025-07-03

**Clarity:** 4
**Significance:** 4
**Originality:** 4
**Rating:** 5
**Confidence:** 5

**Summary:**

This paper proposes an algorithm for reconstructing procedural 3D models (as Blender Python API scripts) from point cloud input. The primary contributions are an expressive Blender Python API library (embedded DSL) capable of some non-trivial CAD operations (primitive creation, beveling, patterning, swept and lofted shapes), a technique for generating large data sets of semantically commented and part segmented programs/models, and a large dataset of such models. The dataset is generated by first randomly synthesizing part-level programs and training a model to reconstruct parts in the embedded DSL from point clouds. The final dataset is constructed by sampling from Infinigen Indoors and using the part-level model to generate DSL code for each part of the sampled models, rejection filtering for reconstruction quality.

**Questions:**

- Was the train/test split done over sampled models or over inifinigen parametric models?
- why were "translation" and "bridge loop" used as names when these operations are traditionally called "sweep" and "loft"? Is there a difference between these and their normal CAD versions?
- Are the part dataset probabilistic programs hand-crafted or somehow sampled from the grammar?
- In Figure 3, the example code seems to show a syntax where the primitive creation operations can be passed arrays of names and parameter lists rather than single names and parameter lists; is this done in-practice in the data set, and if so why is this preferred over separating into multiple lines?

**Ethical Concerns:**

["NO or VERY MINOR ethics concerns only"]

**Final Justification:**

I maintain my evaluation that this paper should be accepted.

The authors addressed my primary concern about generalizability to a sufficient degree. In my response to their rebuttal I suggested a second experiment that could further quantify the generalizability, but I do not think it is critical for the paper to be published.

The 3 changes I would like to see to the manuscript before it is published are:

1. A note in the text that "translation" and "bridge loop" are equivalent to the more common names "sweep" and "loft"
2. An improved description of the parts dataset construction protocol, specifically the collaboration between GPT and human experts. This could be done as an appendix added for reproducibility; the main text only needs the same level-of-detail as they gave in their response "Q3" in the rebuttal.
3. The parameter-array calling convention and its use in the dataset should be described for reproducibility.

**Limitations:**

Limited number of object categories supported.

**Quality:**

4

**Strengths And Weaknesses:**

Strengths:
- Dataset generation method is generalizable to larger procedural modeling contexts
- Reconstruction in most categories appears robust and high quality
- Output is semantically labeled procedural models, good for editability and understanding

Weaknesses:
- Since the test set is construction from the same base set of parametric models (Infigen Indoors) as the training set, it is possible that the model is primarily memorizing the structure of the infinigen procedural models rather than a generalizable pattern. It would be good to see reconstruction experiments performed on non-infinigen inputs.

---

> ### Author Rebuttal · Authors · 2025-07-27
>
> **Thank you very much for your detailed and insightful review!** We greatly appreciate your recognition of the strengths of our work, including the generalizability of the dataset generation method, the robustness of reconstruction, and the value of semantically labeled procedural models for editability. Your valuable suggestions and questions also help us further improve the clarity and rigor of our research. Below we address the weaknesses and questions systematically.
>
> **Response to Weaknesses**
>
> We fully agree with your concern that verifying performance on non-Infinigen inputs is critical for assessing generalization capabilities. To demonstrate our model’s ability to generalize beyond the Infinigen dataset, we have conducted reconstruction experiments using non-Infinigen data. Specifically, we leveraged Trellis[1], a currently popular generative model that takes single images as input and outputs corresponding meshes, to create our experimental data. We collected images of three categories (chairs, shelves, and bottles) from online sources, with 10 images collected for each category. These images were input into Trellis to generate meshes, from which we sampled point clouds to serve as input for MeshLLM. To further demonstrate that our model does not merely memorize the training data, we selected examples from our training dataset that have the smallest CD loss with the point clouds generated by Trellis. All metric results are presented in the table below.
>
> | Method         | Chairs         |              | Shelves        |              | Bottles        |              |
> |----------------|----------------|--------------|----------------|--------------|----------------|--------------|
> |                | CD Loss (↓)    | IoU (↑)      | CD Loss (↓)    | IoU (↑)      | CD Loss (↓)    | IoU (↑)      |
> | PLAD           | 0.09          | 0.13         | 0.12          | 0.16         | 0.21          | 0.22         |
> | Shape2Prog     | 0.03          | 0.46         | 0.06          | 0.06         | 0.25          | 0.40         |
> | Dataset Matches | 0.25          | 0.11         | 0.08          | 0.17         | 0.08          | 0.41         |
> | **MeshLLM**    | **0.005**      | **0.52**     | **0.011**      | **0.61**     | **0.0015**      | **0.78**     |
>
> *Table X: Metrics of different methods on non-Infinigen data.*
> *Note: "Dataset Matches" represent examples from the training set with minimal CD loss to Trellis-generated point clouds.*
>
> This experiment shows that our model maintains satisfactory performance on non-Infinigen data, demonstrating its generalization capability. Additionally, we argue that the model does not merely memorize Infinigen-specific structures. If it did, the metrics of our model’s predictions are unlikely to outperform those of the dataset matches, which represent the closest possible examples from the training set. However, we found that the reconstruction results are suboptimal when attempting to reconstruct objects from categories not present in the training dataset. To further strengthen generalization in future work, we plan to expand to more extensive and diverse datasets such as objaverse[5], and incorporate reinforcement learning techniques (DPO[2], GRPO[3]) into our training pipeline, like DeepMesh[4], to boost the model’s ability to handle broader data distributions.
>
>
> **Response to Q1**
> We generated our dataset using the Infinigen Indoor method. In this approach, each object category is associated with only one procedural generation model. Parameters governing the object shape are sampled and subsequently fed into the model to generate objects. After generating all objects through this process, the entire dataset is split into training and test sets.
>
>
> **Response to Q2**
> We sincerely appreciate you raising this question. Our naming of the data generation methods was referenced from operational conventions in Blender. In fact, "translation" and "bridge loop" are fundamentally identical in principle to the "sweep" and "loft" operations in traditional CAD systems. To achieve terminological consistency with standard CAD practices, we will revise these method names to "sweep" and "loft" in the revised manuscript, and we are grateful for your insight that prompted this important clarification.
>
>
> **Response to Q3**
> The probabilistic programs for the part dataset are developed through a structured collaboration between AI and human expertise. We provided GPT with detailed specifications for each function, including its purpose, input/output parameters, functional scope, and generatable shape types, to generate initial program drafts. Domain experts then reviewed these drafts to assess their quality and ensure coverage of diverse shape variations. For cases where the AI-generated programs were insufficient, human experts manually refined or authored the necessary code to achieve comprehensive coverage. This iterative workflow combines the efficiency of AI-driven generation with human expertise to ensure reliability and diversity in the dataset.
>
>
> **Response to Q4**
> Yes, we have indeed implemented this array-based parameter passing functionality. The code shown in Figure 3 is **extracted directly from our dataset**, with only some parts omitted for brevity. We specifically designed this feature to enable the passing of a list parameter to generate multiple shapes simultaneously. The main purposes of this design are to reduce code length, improve code representation efficiency, and be more resource-friendly for LLM—ultimately reducing training costs and inference time.
>
>
> **References**
> [1] Xiang, J., Lv, Z., Xu, S., Deng, Y., Wang, R., Zhang, B., Chen, D., Tong, X., & Yang, J. (2024). Structured 3D Latents for Scalable and Versatile 3D Generation. arXiv preprint arXiv:2412.01506.
> [2] Rafailov, E., Tang, L., Ziegler, D. M., & Leike, J. (2023). Direct Preference Optimization: Your Language Model is Secretly a Reward Model. arXiv preprint arXiv:2305.18290.
> [3] DeepSeek. (2024). DeepSeekMath: Pushing the Limits of Mathematical Reasoning in Open Language Models. arXiv preprint arXiv:2402.03300.
> [4] Zhao, Ruowen, et al. "Deepmesh: Auto-regressive artist-mesh creation with reinforcement learning." arXiv preprint arXiv:2503.15265 (2025).
> [5] Deitke, Matt, et al. "Objaverse: A universe of annotated 3d objects." Proceedings of the IEEE/CVF conference on computer vision and pattern recognition. 2023.

---

> > ### Comment · Reviewer_WTgQ · 2025-08-04
> >
> > Thank you for addressing my questions and concerns!
> >
> > Q2: Since your model was trained with those names, changing the names in the manuscript does not make sense; I would recommend instead that you put in a note that these are equivalent to loft and sweep operations. I think this is particularly important for "translation" because that operation is used in many systems for a rigid transformation of a whole object rather than an extrusion operation.
> >
> > Q3: Thank you also for including the generalization experiment -- this greatly strengthens the argument for the usefulness of your system. In addition to looking at geometric similarity to the dataset programs, I would also recommend looking at program similarity. This experiment demonstrates that MeshLLM is not directly duplicating dataset models, but it does not differentiate between program lookup and parameter selection (e.g. "fitting" a dataset program to the image) versus construction of new programs. Both capabilities are useful, but the latter is more generalizable. (This experiment is *not* a condition of acceptance for me, I just want to know the answer!)
> >
> > Q4: Does the dataset have a mix of array-based and individual parameters? The existence and semantics of this calling convention should be somewhere in the paper or supplemental for completeness, and if it is the primary syntax used in the dataset and generated by the model then this should be explicitly pointed out in the main paper.
> >
> > Conclusion: I am going to continue to recommend acceptance -- I ask that the authors
> >
> > 1. add a note in the API description that "translation" and "bridge loop" are equivalent to "sweep" and "loft" to aid readers in understanding their meaning
> > 2. As other reviewers noted as well, the protocol for generating the parts dataset was not clear. The explanation in this rebuttal was much clearer; please describe this protocol in the paper or supplement
> > 3. Add the note about the array parameter convention as described above for completeness and reproducibility.

---

> > > ### Author Response · Authors · 2025-08-06
> > >
> > > We sincerely thank the reviewer for your thoughtful feedback and valuable suggestions. We have carefully considered each of the points raised and address them as follows:
> > >
> > > **Q2:** Thank you for the suggestion. We will include a note in the revised version clarifying that the "Translation" operation is equivalent to "Sweep," and the "Bridge Loop" operation is equivalent to "Loft."
> > >
> > > **Q3:** We appreciate the reviewer’s insightful comments, which allow us to further discuss this topic. We address the question from two perspectives:
> > >
> > > - **At the part level:** The code predicted by MeshLLM always consists of functions that exist in our function library; it does not generate entirely new functions.
> > >
> > > - **At the object structure level:** Our model is capable of reconstructing objects with structural configurations that are not present in the training dataset. Here, we define the structure of an object as the set of the number of parts corresponding to different semantic components. For example, the structure of a chair can be defined as a set like {backrest: one, seat: one, legs: four}. Any variation in the presence of different semantic components, or in the number of parts under the same semantic label, constitutes a different object structure.
> > >
> > > **Q4:** Yes, the dataset indeed contains a mix of array-based parameters and individual parameters. When constructing the dataset, we choose the parameter format based on what best fits the object: individual parameters are used for simpler parts, while array-based parameters are adopted when a part contains multiple similar shapes. We will clearly describe this convention in the revised version of the paper.
> > >
> > > Once again, we sincerely thank the reviewer for your constructive feedback. We will make sure to clearly address all of the above-mentioned points in the revised version of the paper.

---

### Decision · Program_Chairs · 2025-09-17

**Decision:**

Accept (poster)

**Comment:**

This submission proposes an object-to-code generator based on a shape tokenizer and an LLM. As part of their contributions, the authors also introduce a comprehensive set of Blender Python APIs and a pipeline to construct a large-scale paired object–code dataset for developing and training the proposed generative model. While the overall framework seems straightforward, the authors demonstrate that the Blender APIs and the large-scale dataset can enable high-quality 3D asset generation through Llama fine-tuning. All reviewers gave positive scores (5, 5, 4, 4) and agreed on acceptance. Hence, the AC recommends poster acceptance.

A reviewer requested three changes for the final version:
1. Add a note in the text clarifying that “translation” and “bridge loop” are equivalent to the more common names “sweep” and “loft.”
2. Provide an improved description of the dataset construction protocol for parts, specifically the collaboration between GPT and human experts. This could be included as an appendix for reproducibility.
3. Describe the parameter-array calling convention and its use in the dataset for reproducibility.